# Learning steers the ontogeny of an efficient hunting sequence in zebrafish larvae

**Konstantinos Lagogiannis\*, Giovanni Diana, Martin P Meyer\***

Centre for Developmental Neurobiology, MRC Center for Neurodevelopmental Disorders, King's College London, London, United Kingdom

**Abstract** Goal-directed behaviors may be poorly coordinated in young animals but, with age and experience, behavior progressively adapts to efficiently exploit the animal's ecological niche. How experience impinges on the developing neural circuits of behavior is an open question. We have conducted a detailed study of the effects of experience on the ontogeny of hunting behavior in larval zebrafish. We report that larvae with prior experience of live prey consume considerably more prey than naive larvae. This is mainly due to increased capture success and a modest increase in hunt rate. We demonstrate that the initial turn to prey and the final capture manoeuvre of the hunting sequence were jointly modified by experience and that modification of these components predicted capture success. Our findings establish an ethologically relevant paradigm in zebrafish for studying how the brain is shaped by experience to drive the ontogeny of efficient behavior.

**\*For correspondence:**
costaslag@gmail.com (KL);
martin.meyer@kcl.ac.uk (MPM)

**Competing interests:** The authors declare that no competing interests exist.

## Introduction

The study of animal ethology has demonstrated an instrumental role of early experience in permanently embedding specific information about the environment in the developing animal, and in extending and enhancing behavioral repertoires (*Purves, 1985*; *Bateson, 1981*).

An example of a dynamic behavior that is found to benefit from learning by experience is predation as, across diverse species, it involves predicting and responding to the behavior of another animal. In altricial species, which require an extended period of parental care, young animals learn to hunt from direct experience, and from mimicking the behavior of parents or other conspecifics (*Danchin et al., 2004*). Even in precocial species, in which hunting behavior is developed prenatally, hunting can nevertheless be modified by experience. For example, with experience, hatching snakes improve in their ability to capture prey (*Mehta, 2009*) and orb-web spiders build more effective webs (*Heiling and Herberstein, 1999*). In fish, the early fine-tuning of development to the available prey types can be critical for survival in the wild. Studies of hatchery-raised fish have found that a lack of prior exposure to prey, before being released into the wild, can strongly reduce their chances of surviving to adulthood (see *Coughlin, 1991*; *Brown et al., 2003*; *Blaxter, 1986*; *Meyer, 1986*; *Dutton, 1992*; *Cox and Pankhurst, 2000*). It is unlikely that exposure to prey simply facilitates certain aspects of behavioral ontogeny that would have developed with time (*Bateson, 1981*), because in at least some fish species, the detection, handling and capturing of prey is enhanced only for the particular type of prey they have experienced (*Meyer, 1986*; *Cox and Pankhurst, 2000*; *Drost, 1987*). Such evidence suggests that the ontogeny of hunting behavior relies on relevant early experience to fully develop (*Brown et al., 2003*; *Warburton, 2003*). Learning by experience could modify both the perceptual and the kinematic components involved in prey detection, pursuit and capture. However, which aspects of hunting behavior are malleable by experience and how these contribute to increasing hunting performance remains largely unknown.

The larval zebrafish is an ideal vertebrate model for the detailed study of behavior and its associated circuits. They have a diverse behavioral repertoire that can be precisely measured (*Orger and de Polavieja, 2017*; *Marques et al., 2018*), and neural activity can be recorded non-invasively and at single-neuron resolution throughout the whole brain (*Ahrens et al., 2013*; *Wolf et al., 2015*; *Portugues et al., 2014*; *Naumann et al., 2016*; *Kawashima et al., 2016*). Larval hunting appears as a distinct behavioral mode that is composed of several component actions chained together into a sequence (*Borla et al., 2002*; *McElligott and O'malley, 2005*; *Trivedi and Bollmann, 2013*). While significant progress has been made in characterizing the kinematics of larval zebrafish hunting in detail (*Bianco et al., 2011*; *Trivedi and Bollmann, 2013*; *Patterson et al., 2013*; *Borla et al., 2002*; *Mearns et al., 2020*), and in describing the circuits and cell types involved in this behavior (*Bianco and Engert, 2015*; *Antinucci et al., 2019*; *Romano et al., 2015*; *Semmelhack et al., 2014*; *Preuss et al., 2014*; *Muto et al., 2017*; *Muto et al., 2013*; *Henriques et al., 2019*), it is not known whether experience affects the ontogeny (*Westphal and O'Malley, 2013*) and effectiveness of their hunting sequences. To address this, we used high-speed imaging and behavioral tracking of freely swimming zebrafish larvae to compare hunting behavior and performance between larvae that have been reared with live prey and those that have not.

Consistent with previous reports, we find that hunting episodes begin with a convergent saccade and a turn, which accounts for a large proportion of the orienting response toward prey. The convergent saccade is maintained throughout the hunting sequence during which larvae home in on the target using a series of temporally discrete swim bouts, which successively minimize the distance to prey, up to a point where they can perform a final capture manoeuvre. We found that experience resulted in a modest increase in the probability of initiating hunting behavior, suggesting that the ability to detect prey or the motivation to hunt may be influenced by prior experience. However, the main effect of experience was a marked increase in the probability of a hunt sequence resulting in successful capture. Detailed examination of successful hunting sequences revealed that experienced larvae were kinematically distinct in at least two aspects. Firstly, at the onset of hunting we found that they make an initial turn-to-prey that undershoots prey azimuth. Paradoxically, inexperienced larvae display initial turns that do not undershoot, but rather align them more with prey azimuth. Secondly, we find that experienced larvae are more likely to employ high-speed capture swims that also tend to be initiated at a longer distance from prey. We show that in experienced larvae capture speed is coordinated with distance to prey, and is also combined with undershooting in their turn-to-prey behavior. We then identify that these coordinated behaviors contribute to the capture success of hunt-sequences, and can be used to predict larval hunting efficiency.

## Results

Zebrafish larvae were reared in one of three different feeding regimes: a live-fed (LF) group, which received Rotifers, a group that was not-fed (NF), and a third group that received dry growth food (DF). After 2 days of feeding, the effects of hunting experience in the LF group were examined. The DF and NF groups served as controls for behavioral effects that can be attributed to nutritional state (see Materials and methods). A graphical timeline of our experimental protocol is shown on *Figure 1*. Consistent with previous observations we did not observe a decrease in survival rates of larvae in the NF group (*Hernandez et al., 2018*). However a small, but statistically significant, effect on growth was determined by comparing means of standard lengths (SL) (*Parichy et al., 2009*) at 7dpf between groups (NF = 4.15 mm, LF = 4.37 mm, DF = 4.21 mm see Appendix 1).

Each larva was left to settle for approximately 1–2 hr before being recorded for 10 min in each test condition: first in a dish that does not contain prey, to observe *spontaneous* behavior, and then in the presence of prey to observe *evoked* hunting behavior (see Materials and methods). Video recordings were then analysed using custom-written tracking software (see Materials and methods). We defined and detected hunt events as blocks of consecutive video frames showing a persistent convergent saccade ($\geq 45°$) (*Bianco et al., 2011*; *Patterson et al., 2013*; *Bianco and Engert, 2015*; *McElligott and O'malley, 2005*). We will refer to the number of detected hunt events as hunt frequency or rate, as all counts were obtained from recording sessions of equal duration (10 min).

For the analysis of behavioral data we developed statistical models of behavior and employed a Bayesian approach (see Materials and methods), which utilizes a combination of data and prior insights to obtain updated (posterior) statistical distributions of model parameters. Unlike less

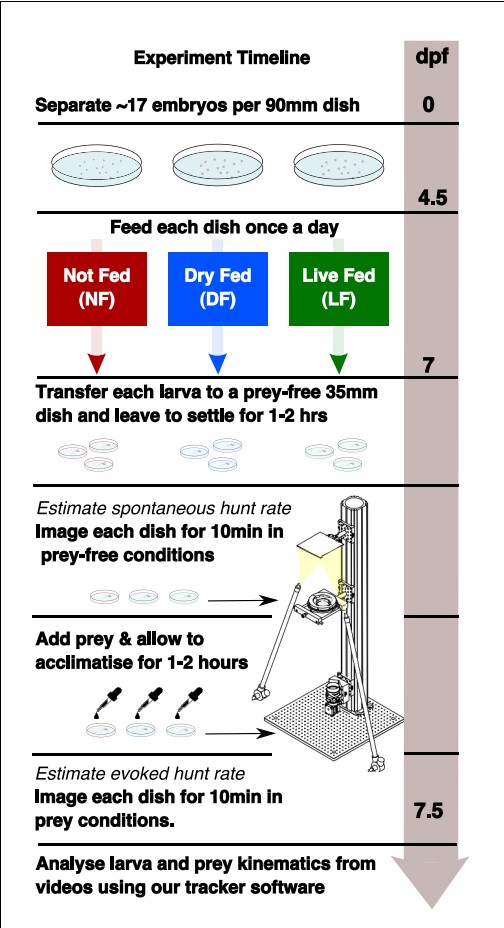

**Figure 1.** Experimental timeline showing the different feeding regimes during rearing and the recording of spontaneous and evoked hunt behavior. Embryos are separated in three dishes, with differential feeding initiated at 4.5 dpf. Each group receives a feed once per day (see Materials and methods). Behavioral recording is performed at 7 dpf. Spontaneous eye-vergence events are measured by recording individual larvae for 10 min in the absence of prey. Live prey ($\approx 30$ Rotifers) are then added. Acclimatization in prey conditions for a period of $\approx 1$–2 hr is followed by recording of larvae in the presence of live prey. Live prey is topped up to initial level prior to recording.

informative methods based on point estimates of model parameters (maximum-likelihood), our approach is based on the analysis of their full posterior distributions, accounting for all sources of noise in behavioral data when comparing different rearing groups. Thus, Bayesian inference provides a rigorous statistical approach to support our conclusions.

## Experience increases hunt rate but decreases hunt duration

Rearing conditions may affect the motivation to hunt and/or the ability to detect prey (*Jordi et al., 2015*; *Filosa et al., 2016*). Although hunting behavior is always accompanied by eye-vergence, eye-vergence may also occur spontaneously in the absence of prey. To account for this, we compared evoked and spontaneous eye-vergence events (referred to as hunt events from now on) across groups. We find that in all groups the evoked hunt-rate is markedly increased compared to the spontaneous hunt rate. *Figure 2A–C* show that the cumulative distribution function (CDF) of evoked hunt frequency is clearly shifted toward higher rates compared to the respective spontaneous hunt frequency CDF of each group. There were examples of larvae, across groups, whose evoked hunt frequency was less than their spontaneous one (*Figure 2D*). This reduction could simply be due to unobserved hunt events that occurred outside the recording system's central region of interest, or it may suggest prey-induced inhibition of hunting behavior in some individuals.

To statistically compare hunt-rates between groups, we modeled the data using a negative binomial distribution (see Materials and methods). The resulting model distributions appear to capture our empirical distributions of hunt-frequency very well; the lines in *Figure 2A–C* represent the CDFs obtained from 200 posterior samples of the negative binomial parameters overlaid with the hunt-frequency data used to fit the model (open squares). The inferred hunt-rate parameter distributions, shown on *Figure 2E*, reveal that the spontaneous hunt-rates between the three groups have similar means ($\mu_{R_s}^{LF} = 3.6$, $\mu_{R_s}^{NF} = 3.8$, $\mu_{R_s}^{DF} = 3.5$), with NF showing spontaneous eye-vergence with slightly higher frequency ($P[\mu_{R_s}^{NF} > \mu_{R_s}^{LF}] = 0.60$, $P[\mu_{R_s}^{NF} > \mu_{R_s}^{DF}] = 0.66$). The evoked hunt rates for all groups are clearly distinct from the spontaneous rates, and the evoked hunt rates are similar in the DF and NF groups ($P[\mu_{R_e}^{NF} > \mu_{R_e}^{DF}] = 0.53$, near chance level). However, the estimated evoked hunt rate distributions show that the LF group hunts with higher frequency than control groups ($P[\mu_{R_e}^{LF} > \mu_{R_e}^{NF}] = 0.71$, $P[\mu_{R_e}^{LF} > \mu_{R_e}^{DF}] = 0.74$), with mean estimated evoked hunt rate for LF being $\mu_{R_e}^{LF} = 13.3$, while these are $\mu_{R_e}^{NF} = 11.9$ and $\mu_{R_e}^{DF} = 12.1$ for control groups. These findings suggest that the ability to detect prey or motivation to hunt may be increased, albeit modestly, by prior experience of live prey. To examine likely differences in motivational state in more detail, we compared hunt durations.

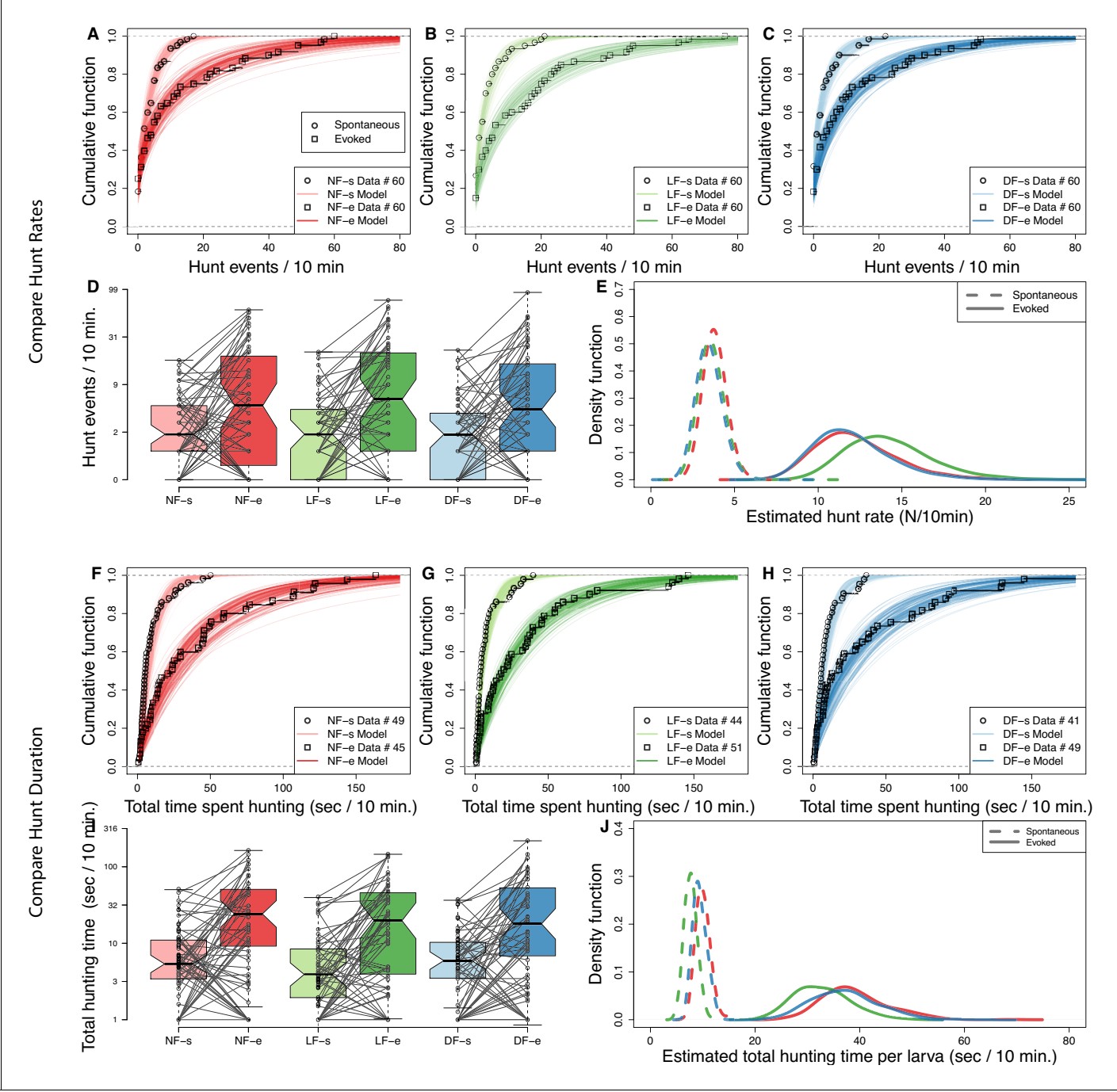

**Figure 2.** Experience increases hunt rate but decreases total time spent hunting. Hunting effort of each group is characterized in terms of the distribution in number of hunt events and total hunting duration recorded from the 10 min behavioral recordings. The two test conditions, in the absence and in the presence of prey, are modeled separately to evaluate *spontaneous* (s) and *evoked* (e) hunt events, respectively. (**A,B,C**) Cumulative density function (CDF) of hunt event counts per larva (open squares) reveals that hunt-event frequency increases across groups once prey is added (NF, not-fed; DR, dry-fed; LF, live-fed). Lines indicate 200 cumulative density functions of negative binomial distributions, which have been inferred from hunt-frequency data. (**D**) Box plots showing number of hunt events per larvae indicate similar spontaneous and evoked counts in each feeding group. Connecting lines indicate that for most larvae the number of hunt events increases on addition of prey. (**E**) The distribution of estimated mean hunt-rate for each group, as inferred from the models' parameters, confirms that hunt-rates increase from spontaneous (dotted lines) to evoked (solid lines) conditions. All groups (indicated by line color) show similar mean spontaneous hunt-rates ($R_s$), with a somewhat higher rate observed in NF larvae ($P[\mu_{R_s}^{NF} > \mu_{R_s}^{LF}] = 0.60$, $P[\mu_{R_s}^{NF} > \mu_{R_s}^{DF}] = 0.66$). In the presence of prey, however, LF larvae are more likely to exhibit higher hunt-rates than NF and DF larvae ($P[\mu_{R_e}^{LF} > \mu_{R_e}^{NF}] = 0.72$, $P[\mu_{R_e}^{LF} > \mu_{R_e}^{DF}] = 0.75$). Overall, the mean estimated group hunt-rate (events/10 min.) in spontaneous/evoked conditions were

*Figure 2 continued on next page*

*Figure 2 continued*

$\mu_R^{LF} = 3.6/14.3$, $\mu_R^{NF} = 3.8/12.3$, $\mu_R^{DF} = 3.5/12.0$. (F,G,H) Cumulative function plots showing the total time spent hunting under spontaneous and evoked conditions. Open squares show recorded data and lines indicate 200 cumulative density functions drawn from similar statistical model as in (A,B,C, see Materials and methods). All groups show an increase in total time spent hunting when prey is added. (I) Box plots showing the amount of time spent hunting increases from spontaneous to evoked test conditions for most larvae. (J) The estimated densities of mean hunt-duration of each group, as inferred from the model, clearly show that on average larvae spent more time hunting in evoked conditions (solid lines) than in spontaneous (dotted lines) conditions. Although DF and NF distributions look identical, the model reveals a noticeable shift towards shorter hunt durations (D) for LF larvae on average in both evoked ($P[\mu_{D_e}^{LF} < \mu_{D_e}^{NF}] = 0.76, P[\mu_{D_e}^{LF} < \mu_{D_e}^{DF}] = 0.72$) and spontaneous conditions ($P[\mu_{D_s}^{LF} < \mu_{D_s}^{NF}] = 0.90, P[\mu_{D_s}^{LF} < \mu_{D_s}^{DF}] = 0.84$).

The online version of this article includes the following figure supplement(s) for figure 2:

**Figure supplement 1.** Hunt episode duration shorter in LF group in both spontaneous and evoked hunt events.
**Figure supplement 2.** Evoked and spontaneous hunt-rates were tested in similar prey density conditions between groups.

We measured the total amount of time spent hunting by individual larvae from each group in each test condition and conducted a similar analysis as above, again utilizing the negative binomial. To estimate hunt duration, we modeled the total number of video frames spent in hunting mode per larva as the sum of statistically independent events (see Materials and methods). The hunt-duration data show that, across rearing groups, the total amount of time most larvae spent hunting is noticeably increased in the presence of prey. This is clearly reflected in data and model CDFs of total time per larvae of *Figure 2F–H*. However, when comparing the inferred hunt-duration distributions between groups, we find that LF larva are more likely to spend less time in hunt-mode in both spontaneous (mean duration per larva (s) $\mu_{D_s}^{LF} = 8.26$, $\mu_{D_s}^{NF} = 10.8$, $\mu_{D_s}^{DF} = 9.8$, with $P[\mu_{D_s}^{LF} < \mu_{D_s}^{NF}] = 0.90$, $P[\mu_{D_s}^{LF} < \mu_{D_s}^{DF}] = 0.84$), and evoked conditions ($\mu_{D_e}^{LF} = 35.5$, $\mu_{D_e}^{NF} = 47$, $\mu_{D_e}^{DF} = 42.5$ (s), with $P[\mu_{D_e}^{LF} < \mu_{D_e}^{NF}] = 0.76, P[\mu_{D_e}^{LF} < \mu_{D_e}^{DF}] = 0.72$), compared to control groups. Therefore, although in the presence of prey LF larvae have higher hunt-rates to controls, they also spent less overall time hunting. These evidence combined suggest that individual hunt episodes in the LF group have become shorter.

To verify this, we examined the duration of all detected hunt events in each condition (699 spontaneous and 2578 evoked), see *Figure 2—figure supplement 1*. Although the distribution of episode duration is rather wide in both spontaneous and evoked test conditions across groups (95% of data between 0.5–5 s), the mean episode duration ($\mu_E$) observed in the LF group is indeed shorter than controls, in both spontaneous ($P[\mu_{E_s}^{LF} < \mu_{E_s}^{NF}] = 0.80$, $P[\mu_{E_s}^{LF} < \mu_{E_s}^{DF}] = 0.75$) and evoked conditions ($P[\mu_{E_e}^{LF} < \mu_{E_e}^{NF}] = 0.65$, $P[\mu_{E_e}^{LF} < \mu_{E_e}^{DF}] = 0.69$).

## Capture success increases with experience

We next examined whether experience affects capture success. The outcome of 1739 hunt events, in which larvae were clearly involved in following prey, were scored while blind to rearing group (see Materials and methods). As shown on *Figure 3A*, the success rate for the LF group was the highest at 32%, with 21% for the NF group, and 18% for the DF. We then proceeded to statistically evaluate the likelihood of this outcome by inferring the joint probability density of successful prey capture and capture-attempts (see Materials and methods). For this, we employed a negative-binomial to model the number of hunt-events with a capture attempt, and combined it with a binomial distribution model in order to infer the probability of capture success (q) by utilizing our labelled data of capture outcomes (Materials and methods). *Figure 3B* shows that the estimated probability of a successful hunting event is clearly increased in the LF group compared to the NF and DF groups. Consistent with the ordering seen in *Figure 3A*, the NF group shows a slightly higher success probability compared to the DF group, yet, due to the wide region of overlap between the two distributions, there is a good chance the two groups have the same overall capture success probability (see figure caption).

We estimated a distribution for the consumption rate per group using the product of hunt-rate and probability of success (*Figure 3C*). Consumption estimates reveal that larvae from the LF group have a mean consumption rate that is almost double that of the NF and DF groups. We find that DF and NF consumption distributions overlap considerably, with mean consumption approximately 1.8 prey/10 min, while LF's consumption distribution is centred at a mean rate of 3.6 prey/10 min. Although both DF and LF groups received nutrition during rearing, DF show the lowest consumption

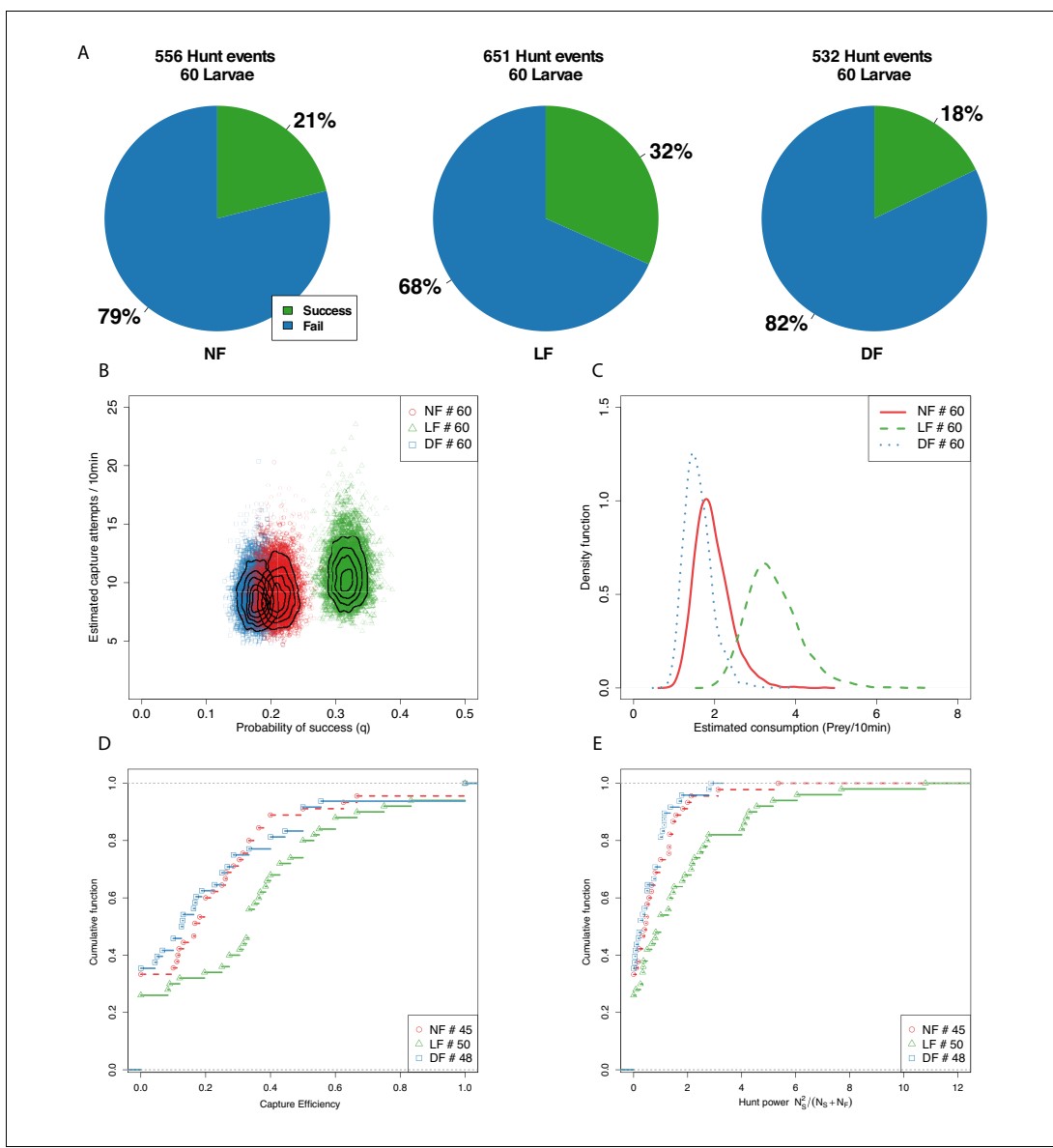

**Figure 3.** Capture efficiency is experience-dependent. (**A**) Proportion of successful and unsuccessful hunt events in each feeding group based on manually labeled outcomes of hunt episodes. Ranking of proportion of successful hunt events: LF>NF>DF. (**B**) We model the joint distribution of probability of capture success and number of capture attempts (see Materials and methods). Each point represents a likely value for the estimated quantities, and contour lines indicate their distribution. The likely distribution of success probability (q) for the LF group has a distinctively higher mean ($P[q^{LF}>q^{NF}] = 1$, $P[q^{LF}>q^{DF}] = 1$) than the NF and DF groups, with $P[q^{NF}>q^{DF}] = 0.55$ confirming that nutrition alone does not explain improved capture success. The distributions for the three groups overlap considerably in terms of capture attempts, but consistent with hunt-rates in *Figure 2E*, the model predicts that mean capture attempts ($\mu_C$) between DF and NF are similar ($P[\mu_C^{NF}>\mu_C^{DF}] = 0.55$), but higher for the LF group ($P[\mu_C^{LF}>\mu_C^{NF}] = 0.74$, $P[\mu_C^{LF}>\mu_C^{DF}] = 0.79$), with differences in mean rates being modest ($\mu_C^{LF} = 10.9$, $\mu_C^{NF} = 9.2$, $\mu_C^{DF} = 8.9$). (**C**) Combining estimated hunt and success rates we plot the probability density function (PDF) of likely consumption per group and find that the LF group's consumption is almost double that of the NF and DF groups. (**D**) CDF of hunt efficiency in terms of fraction of capture successes against capture attempts, shows a general shift rightwards for LF, meaning fewer lower performing larvae as a result of experience. ≈50% of LF larvae have efficiency above 0.33, while that is 0.13 and 0.16 for the DF and NF groups, respectively. (**E**) We define a hunt power index (HPI), as the product of efficiency and number of captured prey, in order to account for the number of captured prey in the scoring of hunting ability (for larvae for which we recorded at least one prey capture

*Figure 3 continued on next page*

*Figure 3 continued*

attempt). A cumulative HPI distribution for each group reveals a difference in slope of LF. A subset of 20% of individuals in the LF group have an HPI higher than top performing DF, NF larvae.

The online version of this article includes the following figure supplement(s) for figure 3:

**Figure supplement 1.** Hunting ability is not explained by larval size.

performance among the three rearing groups. Our finding that both the probability of success and the consumption rate are increased in the LF suggest that prior experience of live prey improves hunting ability. Next, we sought to determine the distribution of hunting-ability among the larvae of each group.

We define capture efficiency as the fraction of capture successes over the total number of capture attempts ($N_{\text{Success}}/N_{\text{Totalattempts}}$) for each larva, and plot the empirical CDF in *Figure 3D*. This distribution suggests that experience has increased consumption rate in the LF group by reducing the number of individuals in the lowest range of hunt efficiency. However, this efficiency measure does not fully reflect hunting performance, because it does not consider the total number of hunt events of each larva, for example a larva succeeding in one out two attempts appears equivalent to a larva that succeeded in 5 out of 10 attempts. For this reason, we defined a hunt power index (HPI) to factor each larva's capture efficiency with the number of captured prey. The empirical CDFs of HPI shown in *Figure 3E* reveal that (20%) of larvae in the LF group have a hunt power that exceeds the highest HPI for larvae in NF and DF groups.

## A fast capture swim is the strategy of success and experienced larvae employ it more often

Selecting the appropriate capture strategy for the type of prey, and executing it with accuracy and precision, will strongly affect a predator's capture efficiency (*Coughlin, 1991*). At the final stage of the hunting sequence, zebrafish larvae execute a capture manoeuvre (see Appendix 3). These capture manoeuvres vary in apparent vigour and distance to the prey from which they are initiated, and can be divided into at least two types based on tail posture and swim speed classification (*Marques et al., 2018*; *Mearns et al., 2020*; *Patterson et al., 2013*). We, therefore, explored whether experience modifies capture strategy and whether this associates with the increased hunting performance of the LF group.

We began by manually classifying capture manoeuvres as either slow or fast, while blind to the rearing group. *Figure 4A–C* reveal that in all three groups the majority of successful hunting episodes involve fast capture swims and that the majority of failed hunting episodes involve slow capture swims. Although slow capture types form the majority of capture swims in all groups, the LF group, which has highest percentage of capture success, also has the largest proportion of high-speed captures (LF≈41% , DF≈21% and NF≈28% ), and the highest ratio of fast captures over slow captures in successful episodes (approximately ×5 LF, ×1.69 DF and ×3 NF). Collectively, these results indicate that successful captures are more likely to result from fast capture swims, and that fast capture swims are employed more frequently and more effectively by larvae in the LF group. However, the above breakdown was based on a subjective estimation of capture speed from which it is difficult to establish reproducible and accurate classification criteria.

For an objective classification of capture swim types, we employed a statistical approach that clustered captures based on their speed and distance from prey (see Materials and methods). Capture speed was measured as the peak speed (mm/s) during the last motion bout in the hunting sequence prior to a successful prey capture ('capture bout'), and prey distance was measured from the tip of the mouth point prior to capture bout initiation (see Materials and methods). The speed and distance data points were then clustered based on a model composed of a mixture of two joint-normal distributions (see Materials and methods). We limited our analysis to successful hunting routines only ($n_{\text{NF}} = 69, n_{\text{LF}} = 92, n_{\text{DF}} = 45$), because in these instances the aims and outcomes of the observed behavior were unambiguously the same. In contrast, comparing between failed hunting sequences would be less straightforward, as failures can occur for many reasons; larvae lose track of the target, abort hunting sequences for no obvious reason, and furthermore, the intended target during failed hunt-sequences is not always obvious.

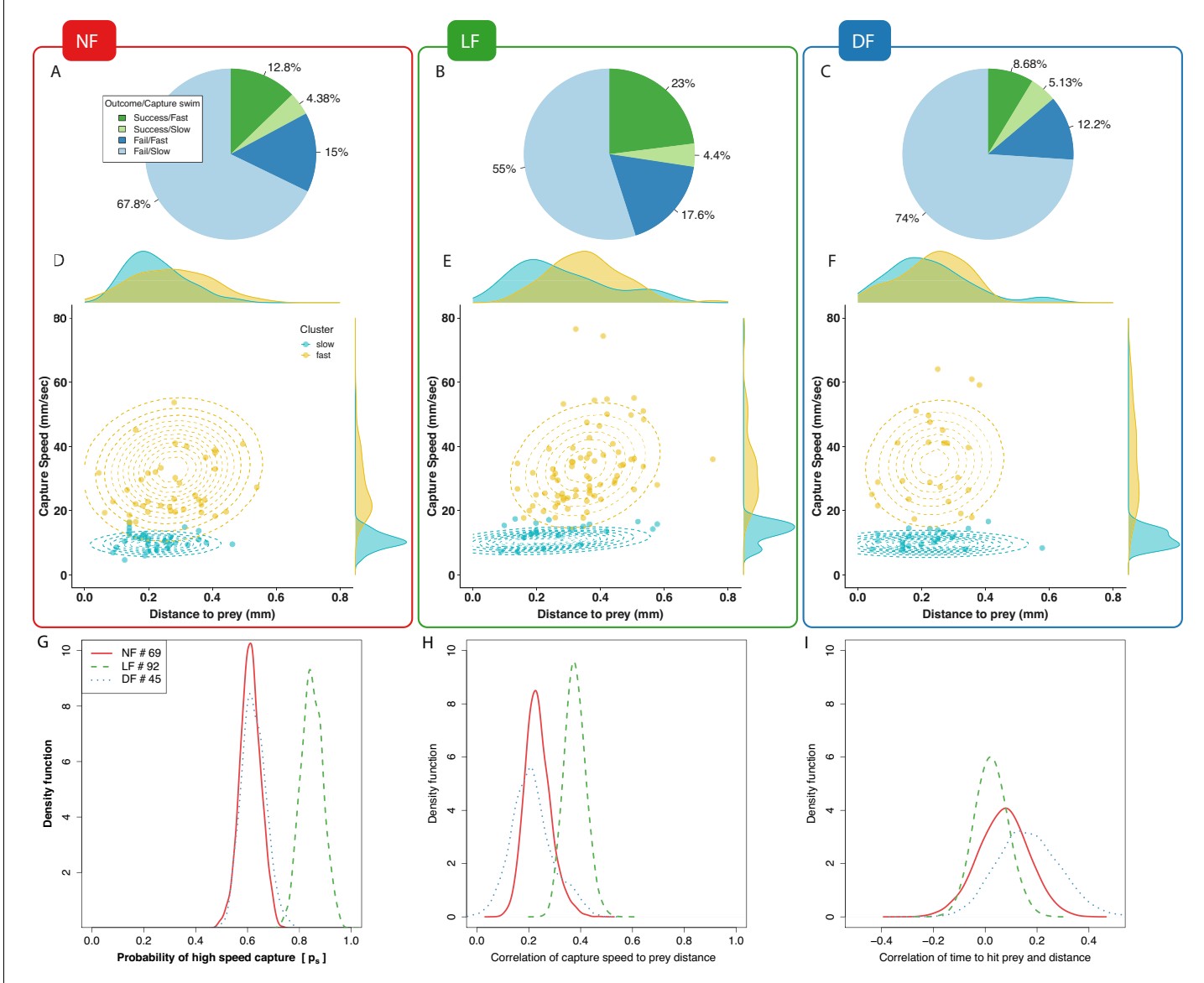

**Figure 4.** Experience-dependent adaptation of capture swims. (**A,B,C**) Manual labeling of captures: Majority of successful hunting episodes used a fast capture swim and these are more frequently observed in LF group. (**D,E,F**) Clustering peak capture-swim speed and prey-distance data points of successful captures in fast/slow (yellow/cyan) using a mixture of two Gaussians model. Marginal density plots for prey-distance show most slower captures executed nearer to prey than fast captures, and that most fast-captures of LF are executed further than those DF,NF. (**G**) Fraction of points that are likely to be clustered as high-speed capture swims ($p_s$) is higher in the LF group than in other groups ($P[p_s^{LF}>p_s^{NF}] = 1$, $P[p_s^{LF}>p_s^{DF}] = 1$), which is in agreement with our labeled results (**A–C**). (**H**) Capture data exhibit an increase of maximum swim-speed with prey-distance. Distribution of Pearson's correlation ($C_{s-d}$) by bootstrapping (80%) capture events pooled across larvae. The mean speed-distance correlation coefficient is positive in capture data across groups (in all groups $p<10^{-3}$ one sample t-test and two sample against shuffled data, see also *Figure 7—figure supplement 4* for a group level statistical model that does not pool data across larvae). (**I**) The correlation between the time it takes to reach prey ($t_{prey}$) and prey-distance in fast-capture swims is smaller in LF compared to NF. Distribution of Spearman's correlation values estimated by bootstrapping (80% of pooled data across larvae) give mean correlations of time vs distance of $\bar{C}_{d-t}^{NF} = 0.07$, $\bar{C}_{d-t}^{LF} = 0.02$, $\bar{C}_{d-t}^{DF} = 0.168$, with the LF group's being the lowest ($\bar{C}_{d-t}^{LF}<\bar{C}_{d-t}^{NF}$ two-sample t-test $p<2.2 \times 10^{-16}$, $P[C_{d-t}^{LF}<C_{d-t}^{NF}] = 0.65$, and $\bar{C}_{d-t}^{LF}<\bar{C}_{d-t}^{DF}$ t-test $p<2.2 \times 10^{-16}$, $P[C_{d-t}^{NF}<C_{d-t}^{DF}] = 0.72$), and with the lowest probability of being positive $P[C_{d-t}^{NF}>0] = 0.76$, $P[C_{d-t}^{LF}>0] = 0.64$, $P[C_{d-t}^{DF}>0] = 0.92$. Scatter plots of $t_{prey}$ are shown on Appendix 4.

The online version of this article includes the following figure supplement(s) for figure 4:

**Figure supplement 1.** The speed of fast-capture manoeuvres also depends on prey distance, and this correlation is strongest for LF data.

**Figure supplement 2.** The relationship between prey-distance and the speed of the capture swim could be a reflection of larvae choosing their capture manoeuvre (fast/slow) depending on prey distance.

The clustering results according to capture type are shown on *Figure 4D–F*, colored according to cluster membership along with the respective contour lines of the joint-normal distribution model for each cluster. In *Figure 4G*, the probability density of a data point being classified as a fast capture swim confirms that fast capture swims are more likely in the LF group than in the control groups. The data also informs the model's mean capture-speed and mean prey distance for each cluster, fast or slow. We find that in general the cluster centers of fast capture swims are located at longer distances from prey than the slow cluster centers, while the fast capture swims of the LF group tend to be initiated further from prey than controls (*Figure 4—figure supplement 1*).

## Capture swim speed and distance to prey become more correlated with experience

We next examined whether distance to prey and capture speed are related and whether this is modified by experience. A relationship between distance to prey and the peak capture speed is apparent in the shape of the fast cluster model, at least for the LF group on *Figure 4E*, (captured in the models' covariances across groups *Figure 4, Figure 4—figure supplement 2A*). To verify this, we measured the Pearson correlation between all prey-distance and capture-speed data points and plotted the distribution of coefficients obtained by repeatedly sampling the correlation ($n = 10^3$) from a random subset of 80% of data points (bootstrapping). *Figure 4H* confirms that a correlation between these variables exists overall and it is stronger in the LF group. Consistently, we find that the covariance of the fast-cluster model also relates capture speed and distance to prey, *Figure 4—figure supplement 2B*. Overall, our findings demonstrate that the frequency of fast capture swims, the distance-to-prey at capture initiation, and the speed-distance correlation increase as a result of rearing with live prey.

For successful capture, the timing of mouth opening needs to be synchronized in relation to prey proximity such that prey enters the mouth cavity either via suction or engulfment (*Drost, 1987*; *Marques et al., 2018*; *Hernández, 2000*; *Coughlin, 1991*). One way of achieving precision in this timing would be to maintain a consistent distance to prey from where to execute capture swims (*Coughlin, 1991*). Alternatively, the capture speed could be adjusted with prey distance in an effort to maintain the timing from capture initiation to reaching prey constant. To evaluate this hypothesis, we calculated distributions of Spearman correlation coefficients, by bootstrapping on 80% of the data points, to reveal if there is relationship between the observed time to reach prey and the prey distance, while preventing correlations being dominated by the absolute scale of the variables. *Figure 4I* suggests that time to reach prey does not vary with distance travelled during the capture swim, and this is more clearly demonstrated in the LF group's fast capture swims. Thus, the ability to adjust capture speed as a function of prey distance is modified by experience. Next, we examined whether the accuracy with which larvae re-orient toward prey during pursuit may also contribute to the LF group's hunting efficiency.

## An off-axis approach strategy develops through experience

A large turn that re-orients a larvae toward prey commonly occurs at the beginning of the hunting sequence (*Bianco et al., 2011*; *Trivedi and Bollmann, 2013*; *Patterson et al., 2013*) (see *Figure 5A*; Appendix 3). This initial turn is proportional to prey azimuth (*Patterson et al., 2013*; *Trivedi and Bollmann, 2013*). We examined if the accuracy of this response is modified with experience, limiting our analysis to successful hunting for the reasons outlined above.

We first established that prey detection occurred over similar angles across the three groups prior to the initial turn. The distributions of the initial prey-azimuths in successful hunt episodes appear bimodal (see *Figure 5—figure supplement 1*), with prey detection from all groups being more likely to occur in response to prey located 35°-50° on either side of the midsaggital axis. A scatter plot of turn response to the initial prey azimuth suggests that these two quantities are proportional (*Figure 5*). The ratio of these two quantities ('turn-ratio') is unity when larvae accurately re-orient toward prey. We found that the data pooled from recorded hunt episodes across larvae gave a mean turn-ratio (μ± SEM) $0.80 \pm 0.029$ for LF group, and this was different to DF $0.98 \pm 0.05$, and NF $0.93 \pm 0.03$ (see *Figure 5—figure supplement 2*). To statistically evaluate potential differences we fitted a linear model and compared turn-ratios based on the slope parameter distributions inferred from the turn data of each group (see Materials and methods).

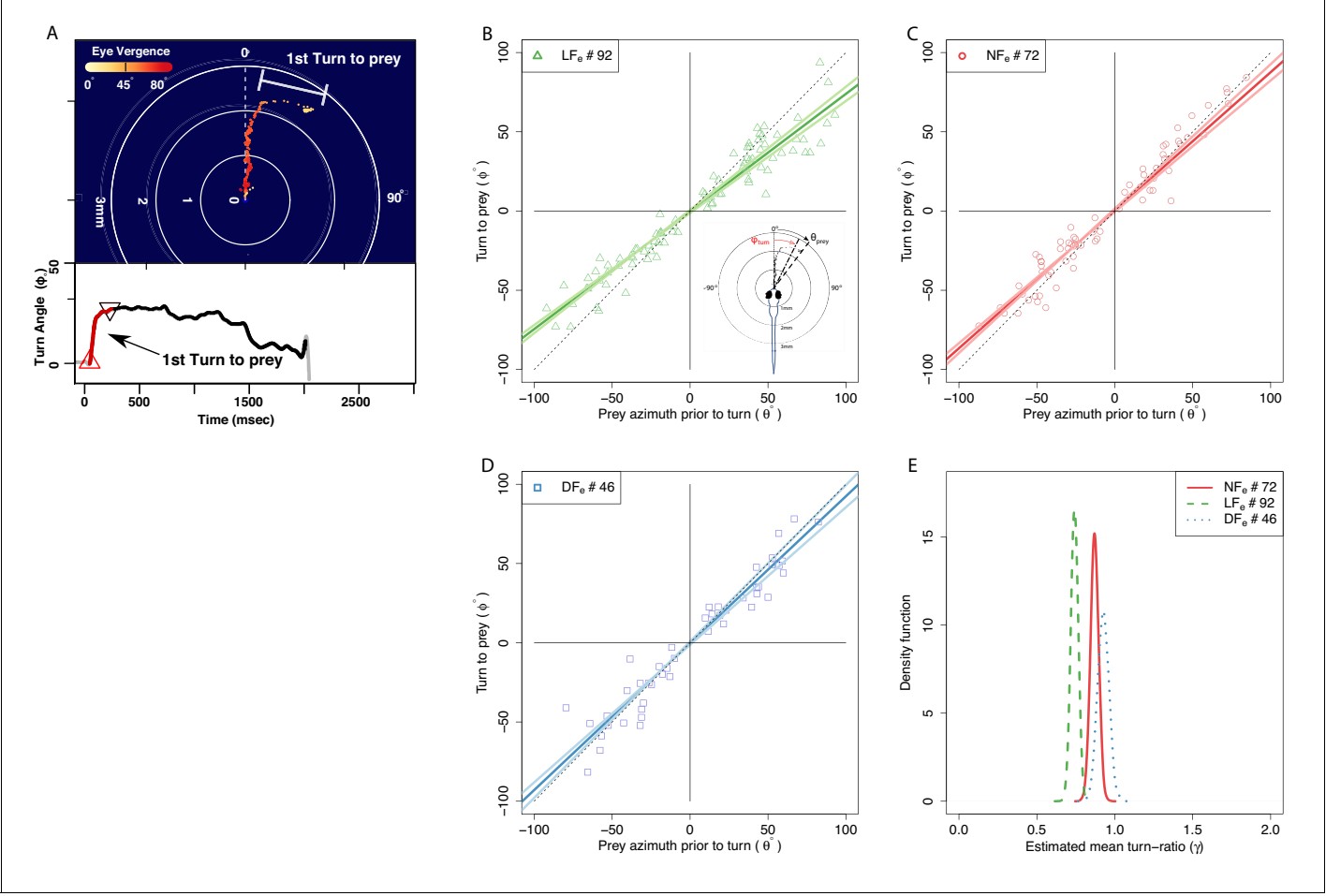

**Figure 5.** The angle of first turn-to-prey is experience-dependent. (**A**) Hunting sequences begin with eye-vergence and a turn bout that re-orients larvae toward prey. We isolate the initial re-orientation toward prey as shown in an example hunting trajectory, which is drawn relative to the larva's heading and mouth position (center), and color coded corresponding to eye-vergence. A large amplitude turn that re-orients it toward prey coincides with an increase in eye-vergence angle. Inset shows turning over time, with red highlighting the extracted first-turn behavior and light grey indicating post-capture turn. (**B–D**) Prey azimuth vs magnitude of re-orienting first-turn data points along with regression lines (dark), and 5–95% confidence intervals (light), according to a linear fit. Hunt events from the LF group show the highest deviation from dotted line, which indicate the slope of turn angles that would precisely align larvae with prey. (**E**) The inferred slope density from the linear statistical regression model reveals that the mean first-turn behavior (γ) of hunt events pooled from the LF group have a lower slope than NF ($P[\gamma^{LF} < \gamma^{NF}] = 1$), and DF ($P[\gamma^{LF} < \gamma^{DF}] = 1$), while DF shows the least undershooting ($P[\gamma^{NF} < \gamma^{DF}] = 0.90$). The ranking in mean turn-ratios according to pooled data across individuals goes LF>NF>DF, but this can be biased toward more active individuals within each group, see *Figure 7* where group behavior is estimated without this bias.

The online version of this article includes the following figure supplement(s) for figure 5:

**Figure supplement 1.** The distributions of prey azimuth at the onset of successful hunting episodes are similar between groups, and favor prey on the lateral visual field.

**Figure supplement 2.** Turn-ratio histogram per group also show undershoot bias for LF.

The distributions of likely turn-ratios estimated by the model shown on *Figure 5E* confirm that the initial turns recorded in the LF group are distinct in undershooting prey azimuth; the slope of the regression model gives a mean 0.73 for LF, 0.87 NF, and 0.92 for DF. This undershooting behavior recorded from the LF group is consistent with previous reports on the first-turn-to-prey of larvae that had also been reared with live prey (Paramecia) (*Trivedi and Bollmann, 2013*; *Patterson et al., 2013*; *Bolton et al., 2019*). Paradoxically, hunt events from the NF and DF groups display initial turns that do not consistently undershoot, but rather align larvae closer to the prey's azimuth. These findings suggest that undershooting on the first turn to prey is a kinematic adaptation of the hunting sequence that develops from rearing with live prey. We next examined whether experience is

required for linking the initial turn behavior with fast capture swims, which were more frequent in the LF group.

## Experience combines off-axis approach and fast capture swim strategies

The LF group exhibits a higher frequency of fast capture swims and a stronger tendency to undershoot in the initial turn-to-prey than controls, but whether these two behaviors are combined within individual hunt events remains unknown. Fast capture swims and undershooting could be used independently of one another or they could be linked either via a learned or innate association. In the case of an innate relationship, which is immutable by experience, undershooting would lead to fast-capture swims regardless of rearing group. Alternatively, if the association between these two behaviors can be modified by experience, the combination of undershoot and fast-capture swim should be unique to hunt events of the LF group.

The relationships between turn-ratio and capture-speed for each successful hunt episode are shown on *Figure 6A–C*, along with densities for each of the two clusters (fast/slow) on the plot margins. An association between fast capture swims and turn ratio would manifest as a leftward or rightward shift in the densities shown along each plot's top margin. A leftward shift, indicating undershooting (turn-ratio < 1), is visible in the LF group's turn-ratio density of the fast capture swims (yellow) on *Figure 6B* while no obvious relationship between turn-ratio and capture speed is seen in the fast capture swims of NF and DF groups *Figure 6A and C*. Thus, inspection of fast-cluster datapoints from successful capture episodes suggests the existence of a bias toward one side of turn-ratio unique to the LF group.

To evaluate the evidence for a correlation between turn-ratio and capture speed, we calculated densities of their correlation by taking multiple random subsamples (80%) from all successful hunting data for each group and calculating Spearman's correlation coefficients for each subsample. The estimated correlation coefficient densities shown on *Figure 6D* evidently confirm that undershooting and capture-speed are correlated in the hunt events of the LF group, while no such clear relationship exists for the control groups' hunt-events. Thus, experience drives the development of correlations between undershooting and fast-capture swims, which suggests that experience is not confined to modifying components of the hunt sequence independently but is also able to combine them.

## Typical hunting behavior is distinct in experienced larvae

Our analysis so far has found that the experienced group's hunting routines are different to control groups' in terms of turn-to-prey and final capture swim. However, these observations may not be representative of typical larval behavior within each group, since the majority of successful hunt-events analysed could be originating from an atypical minority of larvae. In this section, we go beyond comparing pooled hunt-episodes and characterize the prevalent hunting behavior of each group by aggregating behavioral estimates of its larvae. For this, we built a Bayesian hierarchical statistical model (Materials and methods) that provides posterior estimates for the distribution of mean prey-capture behavior for each group of 60 larvae, based on available observations from the subset of larvae that executed successful prey captures in each group.

Consistent with our earlier analysis, we compare mean capture behavior between groups based turn-ratio, capture-speed and distance to prey. *Figure 7A* shows a 3D rendering of the resulting posterior parameter distributions per group, according to which the LF group (green) has higher overall capture speed and distance, and lower turn ratio (see *Figure 7—figure supplement 1* for a more detailed view of parameter space). This result is consistent with our earlier analysis of pooled hunt events (*Figures 4* and *5*). Nevertheless, with this group model, we can confirm that experience has indeed affected the hunting behavior of the larval population and we can also obtain the estimated mean hunting behavior of each larva.

## Capture efficiency is predicted by the combination of off-axis approach and fast capture swims

Next, we provide evidence that the adaptations observed in the hunting routines of the experience group can partly explain their increase in capture success. For this analysis, we only included larvae for which we had at least five capture attempts in order to reduce the error in estimating the mean

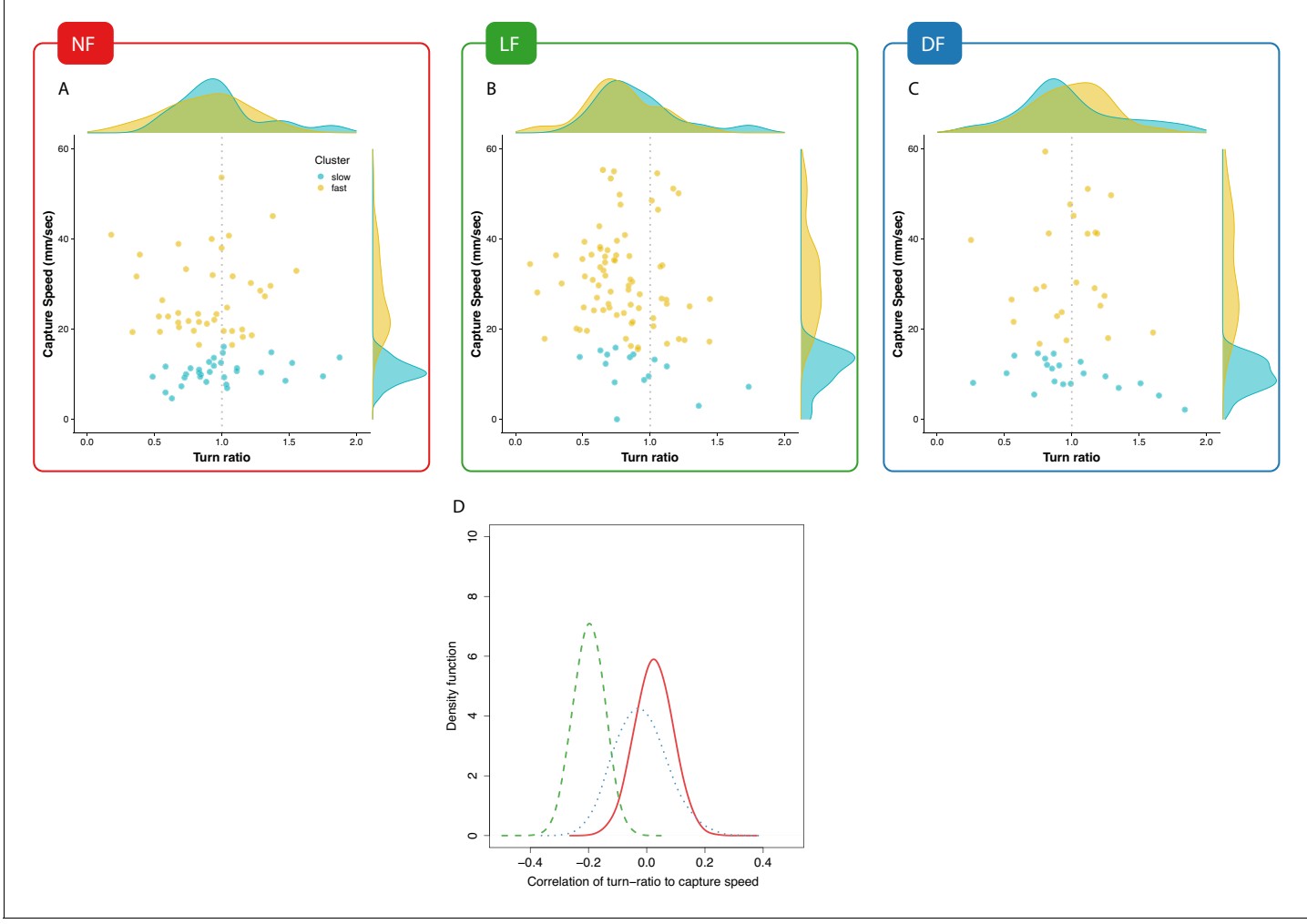

**Figure 6.** Experience associates undershooting to fast capture swims. (**A,B,C**) Pooled data points across larvae on capture-speed and turn-ratio. The densities shown along the top margins suggest that fast captures (yellow, see **Figure 4**) are biased toward low turn-ratios (undershooting) in the LF group, while no such preference is seen in the NF and DF groups. (**D**) Distribution of bootstrapped (80%) Spearman's correlation coefficient between turn-ratio and capture-speed; Hunt events from LF larvae show a relationship between low turn-ratio (undershoot on their first-turn to prey) with the speed of the final capture swim ($p<2.2 \times 10^{-16}$ one-sided t-test, $P[C_{\gamma-S}^{LF}<0]>0.99$). This relationship is not innate but rather driven by experience because although DF and NF larvae may occasionally undershoot on their first-turn to prey, this turn is not systematically associated with a fast-capture swim. NF shows the opposite, a positive correlation ($p<2.2 \times 10^{-16}$ one-sided t-test, $P[C_{\gamma-S}^{NF}<0] = 0.35$), while DF also show evidence of combining undershoot with capture ($p<2.2 \times 10^{-16}$ one-sided t-test, $P[C_{\gamma-S}^{DF}<0] = 0.60$), but correlation is not as strong as in LF ($p<2.2 \times 10^{-16}$ two sample one-sided t-test). The online version of this article includes the following figure supplement(s) for figure 6:

**Figure supplement 1.** Prey azimuth prior to capture manoeuver.

capture efficiency per larva. We generated a six-dimensional vector for each individual that was composed of the three model-estimated values (capture-speed, capture-distance, turn-ratio), which we have used so far to characterize hunt-behavior, along with its estimated capture efficiency, the number of capture attempts, and the empirical mean time to reach prey ($t_{\text{prey}}$) during capture. These behavioral vectors were then combined into a matrix in order to examine the relationship between larvae of different groups using principal component analysis (PCA). In **Figure 7B**, we show the multidimensional space defined by six hunting variables as viewed from the reduced space of the first and second principle components (PC). These two PCs capture approximately 62% of the total variance and approximately 50% of variance in capture efficiency. Consistent with our group model in **Figure 7A**, we find that LF larvae occupy a distinct position in this space that separates them from the control group NF/DF larvae. The LF region is pointed to by axes of capture speed, prey distance

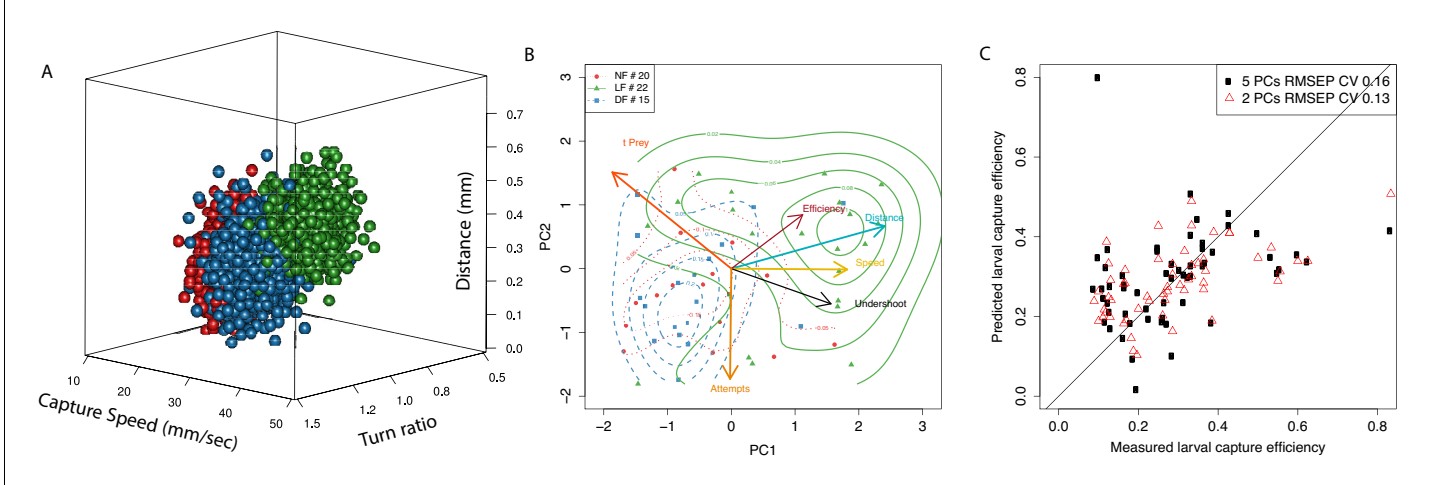

**Figure 7.** Typical behavior of experienced larvae is distinct and relates to their increased capture efficiency. (A) Estimated distribution of mean hunting behavior based on capture speed, distance to prey and initial turn for each group according to a multivariate normal hierarchical model of group behavior. The LF group's model has a mean capture distance ($\mu_d^{LF}$) in successful hunt episodes that is longer than control groups ($P[\mu_d^{LF} > \mu_d^{NF}] = 0.87$, $P[\mu_d^{LF} > \mu_d^{DF}] = 0.82$). Estimated group mean capture distances (mm) are $\mu_d^{LF} = 0.35$, $\mu_d^{NFF} = 0.23$, $\mu_d^{DF} = 0.24$. LF also has the highest mean capture-speed: $\mu_s^{LF} = 28.2$, $\mu_s^{NF} = 17.5$, $\mu_s^{DF} = 23.3$ ($P[\mu_s^{LF} > \mu_s^{NF}] > 0.99$, $P[\mu_s^{LF} > \mu_s^{DF}] = 0.90$). The LF mean turn-ratio $\mu_\gamma^{LF} = 0.85$ shows the group clearly undershoots ($P[\mu_\gamma^{LF} < 1] = 0.96$), as a result of experience as it stands distinct to controls ($P[\mu_\gamma^{LF} < \mu_\gamma^{NF}] = 0.80$, $P[\mu_\gamma^{LF} < \mu_\gamma^{DF}] = 0.70$), whose turn-ratios stand closer to unity $\mu_\gamma^{NF} = 0.94$, $\mu_\gamma^{DF} = 0.92$. (B) PCA in larval hunting behavioral space showing the axes for hunting behavior and the position of individual larva across groups as seen from the first 2 PCs. Larvae are positioned in this PCA space based on their mean hunt behavior (capture speed, turn-ratio, capture distance) as estimated by the model, their mean capture efficiency (min. five capture attempts), along with their number of capture attempts and the mean time to hit prey ($t_{\mathrm{Prey}}$). Distributions per group (contour lines) confirm separation of the LF group from controls and shift toward region pointed to by the efficiency axis (red). Members of the LF group are shown to be more efficient hunters, they undershoot on their first turn to prey (black), and execute fast capture swims (yellow) from a relatively longer distance to prey (cyan), thus suggesting a relationship of these behaviors with hunting efficiency (red). (C) Linear regression model shows that two principal components of a larva's mean behavior, as in B, can be used to predict its capture efficiency.

The online version of this article includes the following figure supplement(s) for figure 7:

**Figure supplement 1.** Detailed sections of 3D parameter distributions shown on panel A.
**Figure supplement 2.** Model-estimated group correlation of capture speed and turn-ratio show LF more likely to combine undershoot with a fast capture swims.
**Figure supplement 3.** Model-estimated group correlation between capture distance and turn-ratio show LF more likely to undershoot with a longer range capture swims.
**Figure supplement 4.** Model-estimated correlation of capture speed and prey distance shows fed groups (LF/DF) more likely to modulate capture speed with distance.
**Figure supplement 5.** PCA applied to mean larva behavior calculated empirically showing agreement with model based in B.

and undershoot, as well as increasing efficiency; larvae from control groups occupy the opposite region in this behavioral space which is in the direction of lower efficiency.

This coarse alignment of efficiency with the axes for capture-speed, distance and undershoot indicates that hunting efficiency may be related to, and possibly predicted by these specific hunting behaviors. To examine this further, we employed principal component regression (*Wehrens and Mevik, 2007*) and assessed the prediction capability of linear models that could utilize any combination of PCs as factors. The maximum cumulative percentage of efficiency variance that can be explained with two PCs, given the factors we used on *Figure 7B*, was found to be ≈32%, and the coefficient of variation (CV) in the root mean squared prediction error (RMSEP) was $CV = 0.13$. In comparison, using all five PCs increased the efficiency variance explained to ≈40%, but also the RMSEP CV = 0.16. The ability of these PC efficiency prediction models are shown on *Figure 7C*.

We have therefore obtained evidence that capture success relates to, and can be partly predicted by utilizing covariates of hunting behavioral measurements within the first 2 PCs. Additionally, we found that these 2 PC are loaded with combined behaviors (as in *Figure 7B*), revealing that

increasing efficiency relies on adapting behaviors in a coordinated manner, as is the case for example between capture-speed and prey distance.

## Discussion

We have shown that first-feeding zebrafish can improve their hunting performance by engaging in their natural hunting behavior. This can be observed as early as 7 dpf, and to our knowledge, it is the first time experience has been shown to affect hunting ability at that early developmental stage. Our experimental design allowed us to examine the comparative effects of different rearing diets on hunting behavior, and disentangle the effects of growth and nutrition from those of experience. By utilizing statistical inference to characterize the effects of rearing diet on the hunt-effort and capture success of each group, we were are able to provide the first comprehensive evidence that differences in consumption are mostly due to experience-depended changes in hunting skill (see *Figure 3D*). We then focused on identifying the behavioral aspects that lead to increased capture efficiency. Key aspects of behavior were extracted from detailed kinematic analysis of hunting sequences, which were then used to build a hierarchical statistical model of mean hunting behavior per group based on the estimated mean behavior of its larva. Using these estimates of behavior, we were able to associates capture success to coordinated adaptations of the hunting sequence by showing that these identified aspects of hunting behavior can be used to predict hunting efficiency.

### Experience-dependent changes in hunt rate, duration and efficiency

Because all hunting sequences in larval zebrafish begin with eye vergence (*Patterson et al., 2013*; *Bianco et al., 2011*; *Bianco and Engert, 2015*), we could use the rate and duration of eye vergence events to estimate hunt motivation and the ability to perceive and respond to prey. We found that the rates and duration of spontaneous and evoked hunt events were very similar in the DF and NF groups. However, compared to the two control groups, there was a modest increase of hunt rates in the LF group only in the evoked conditions. This finding is consistent with previous observations that zebrafish larvae raised with Paramecia performed prey capture behaviors more frequently than those that had not (*Patterson et al., 2013*), and suggest that prior experience may condition prey response, which would be in agreement with findings in other fish species (see *Winfield and Townsend, 1988*).

However, our finding that hunt rates are not elevated in the NF group is seemingly at odds with studies showing that starved larvae are more likely to shift behavioral decisions from avoidance to approach (*Filosa et al., 2016*) and increase their food intake compared to fed larvae (*Jordi et al., 2015*). This apparent discrepancy can be reconciled by the fact that we allow all feeding groups to acclimatize with live prey for 1-2hr prior to recording in the evoked conditions. This time period has been shown to be sufficient to overcome motivational differences between starved and fed larva, as within 40 min differences in the bout patterns expressed between these groups become undetectable (*Johnson et al., 2020*).

Despite the increase in their hunt rate, larvae in the LF group tend to spend overall less total time hunting than larvae in the control groups. This decrease is noticeable in both, spontaneous and evoked conditions, and thus it may not depend on the presence of prey. Although, it is still likely that experienced larvae become faster at catching prey and therefore more time-efficient, the evidence here suggests a change in an internal timing process which is responsible for sustaining the state of hunt activity. Indeed, the distributions for the duration of individual hunt episodes are surprisingly similar between the evoked and spontaneous events of each group (*Figure 2—figure supplement 1*), especially so in the naive group (NF).

We suggest that an intrinsic timer process, similar to the stochastic-state switching mechanism recently proposed to underlie the switching of foraging state between exploration and exploitation (*Marques et al., 2020*), is likely controlling hunt-episode duration, and allows for these to be modulated by factors such as prey stimuli, satiety (*Johnson et al., 2020*) and experience.

We then extended our statistical model of hunt-rates to include the probability of success of each of the evoked hunt events. This more comprehensive model revealed that the estimated mean prey-consumption rate for the experienced group is approximately double that of control groups. This increase in consumption was largely due to improved capture success rather than increased hunt-rates, because the probability of a successful capture was significantly elevated in larvae from the LF

group by ≈ 50% compared to NF larvae, while the difference in the number of capture attempts between the LF and NF groups was more modest (≈ 18%) (see *Figure 3B*). Thus, the effects of experience on hunting behavior are mostly seen as an increase in capture efficiency at this stage of development (7 dpf). The superior capture efficiency of LF against the NF group is not explained by differences in growth, because larvae in the nutrition control group (DF), which tend to be larger than larvae in the NF group (see Appendix 1 and *Figure 3—figure supplement 1*), have the lowest capture efficiency of the three groups. Therefore, the most likely explanation is that superior capture efficiency is caused by changes to the hunting routine.

## Experience-dependent modification of the first-turn-to prey and final capture swims contribute to capture efficiency

There may be multiple paths to success, but some can be more efficient than others. Accordingly, even when comparing hunting episodes that end up in successful captures, there may be differences in hunting behavior between efficient and non-efficient hunters. Indeed, by comparing behavior during successful hunting episodes alone, we were able to unambiguously identify differences in components of the hunting sequence between experienced and control larvae.

We show that hunting sequences of experienced larvae are distinct in at least two respects. Firstly, on detection of prey, larvae from all groups make an initial turn that accounts for a large part of the reorienting response. However, experienced larvae make a first turn that tends to undershoot prey azimuth, whereas larvae in the two control groups make an initial turn that brings them closer to aligning with prey azimuth. Secondly, we find that experienced larvae are more likely to employ high speed capture swims and that these are initiated at greater distances from prey than the control groups. Capture speed and distance are correlated, yet this correlation is highest in the hunt events of the LF group (*Figure 4H*), implying that experience is necessary for tuning the vigour of the capture swim as a function of prey distance. To verify that these behaviors are characteristic of experienced larvae, we embedded larvae of all groups in a principal component space of hunting behavior and found that larvae from the LF group are segregated against the NF and DF larvae, while NF and DF strongly overlapped (*Figure 7B*).

Using only just two principal components we able to build a linear model that predicts larval hunting efficiency, in terms of the fraction of capture successes over total capture attempts, and therefore show that the identified hunting behaviors are related to the probability of capture success (*Figure 7C*). Neither of the two PCs aligned with a particular hunt behavior in isolation, but rather each component fused a mixture of first-turn-to-prey, capture speed, capture distance and number of capture attempts. However, why does modification of capture speed or an undershoot turn-ratio increase hunting efficiency?

## Fast captures initiated from a distance reduce capture failures

Capture probability can vary with prey type as differences in their motion and escape patterns are likely to require different capture strategies (*Drenner et al., 1978*; *Meng and Orsi, 1991*; *Coughlin, 1991*; *Westphal and O'Malley, 2013*). For some evasive prey species utilizing a fast capture speed may be important (*Drost, 1987*; *Westphal and O'Malley, 2013*). There is already evidence that Atlantic salmon, carp (*Cyprinus earpio*) and pike (*Esox IUC US*) larvae improve their capture success by using fast capture-speeds (*Coughlin, 1991*; *Drost, 1987*). Additionally, attacking prey from a distance can minimize the chance of being detected by prey and evoking escape responses. For example some copepods respond to moving objects at a distance (0.5 mm–1 mm) by sensing water disturbances (*Lapesa et al., 2002*). Thus, it would be beneficial to adapt the capture strategy to the patterns of prey motion and escape responses.

However, in this study we used Rotifers, which, unlike other prey types (*Drenner et al., 1978*; *Drost, 1987*), they are not known to exercise any effective evasive response to imminent predator capture strikes (*Coughlin, 1994*; *Lapesa et al., 2002*; *Yúfera et al., 2005*). Adult Rotifers, can move at 0.5–1.5 mm/sec (≈ 1.5–2 body lengths), depending on their size among other factors (*Yúfera et al., 2005*; *Lapesa et al., 2002*), which is in agreement with our measured estimates of mean prey speeds (see Appendix 5).

Naturally, prey speed will affect the rate at which prey azimuth changes from the larva's point of view, and thus the aiming accuracy of the capture manoeuvre may be compromised by moving prey.

The size of the effect that continuous prey motion can have on aiming accuracy can be estimated by calculating the change in prey azimuth during the inter-bout interval preceding the capture swim, which will be taken here to be ≈200 ms (*Trivedi and Bollmann, 2013*; *Johnson et al., 2020*; *Bolton et al., 2019*). A simple geometric examination reveals that changes in prey azimuth due to prey motion scale with inverse of prey distance (see analysis in Appendix 5). Prey traveling in proximity of 0.2 mm to the larva can cause deviations in azimuth that are very likely to affect capture success because they will move prey out of the optimal strike-zone range (±25°), observed across successful capture episodes of all groups (*Figure 6—figure supplement 1*, see also *Mearns et al., 2020*; *Bolton et al., 2019*). Although there is evidence that the larval prey tracking algorithm utilizes prey-speed to bias the heading of the next bout towards the predicted azimuth at a future prey position (*Bolton et al., 2019*), these estimates may also be compromised over short distances. A linear prediction model for prey azimuth, which utilizes perfect prey-displacement estimates, would still suffer significant prediction errors when attempting to capture fast-moving prey in proximity closer than 0.2 mm (see *Appendix 5—figure 1B*). Therefore, an increase in capture distance can lead to improvements in capture success rates because it effectively diminishes errors in aiming caused by prey motion, especially against faster moving prey.

Prey capture from a distance is associated with a kinematically distinct fast-capture manoeuvre (*Bianco et al., 2011*; *McElligott and O'malley, 2005*; *Patterson et al., 2013*; *Borla et al., 2002*; *Mearns et al., 2020*), during which the larva lunges forward reaching the prey in less than 100 ms (see Appendix 4). During this short interval, larvae need to adjust the timing of the gape and suction action to occur right in front of the prey (*Coughlin, 1991*; *Marques et al., 2018*), as otherwise, capture failures occur because the larvae hit and push the prey away (*Coughlin, 1991*). Indeed, the distribution for the time to max-gape reported in *Mearns et al., 2020* is in agreement with our distributions for the time to reach prey in successful captures (see timing densities in Appendix 4), thus supporting the idea that timing of gape is coordinated against the time larvae reach the prey. Since the gape-cycle timing is reportedly stereotyped (*Hernández, 2000*), we therefore suggest that larva zebrafish adjust their swim-speed as a function of prey distance in order to coordinate the time of reaching prey against their max-gape timing. In agreement with this, we find that the time to reach prey is independent of capture distance (see *Figure 4I*), especially in the experienced group (LF), where we also find a stronger correlation between capture distance and speed than controls. Therefore, we contend the coordination of capture-speed and distance develops by experience to achieve correct timing of the mouth opening such that it occurs in proximity to prey as the larva lunges forward. However, how can larva perceive distance to prey, and know they are within capture distance?

## Turn undershoot as an active sensing strategy

We focused our analysis of turn behavior only on the initial turn towards prey because it amounts to the largest re-orientation following prey detection and so it can be representative of the ability to aim toward prey while being less prone to measurement error. Subsequent turn behavior follows the same pattern of undershooting prey azimuth (*Bolton et al., 2019*), and thus we can take the first turn behavior to be typical of a larva's re-orienting behavior. We hypothesize that this undershooting behavior is an off-axis approach strategy that enhances capture success by actively improving the perception of prey distance.

The possession of a distance perception faculty is manifest across all phases of hunting behavior. Prey proximity predicts prey choice when hunting is initiated in the presence of multiple targets (*Bolton et al., 2019*), while the length and intervals of the swim bouts executed during prey approach progressively shorten with prey-distance (*Trivedi and Bollmann, 2013*; *Bolton et al., 2019*; *Johnson et al., 2020*), eventually leading to stopping at a distance appropriate for a capture strike, the success of which, as we discussed in the previous section, also appears to dependent on prey-distance. Although image size is an important feature for triggering hunting in zebrafish larvae (*Bianco et al., 2011*; *Romano et al., 2015*), distance perception through image size may not be sufficient, because vision in zebrafish is monocular with a fixed focus and so retinal image size is an ambiguous cue for the true size of an object.

A potential way to resolve this ambiguity is to utilize cues arising from a target's velocity, as this will depend on observer's distance like in the well-known motion-parallax phenomenon. Although prey velocity cues obtained passively during inter-bout intervals are likely used (*Trivedi and*

*Bollmann, 2013*), these would provide ambiguous distance estimates because prey-speed can vary significantly even among prey of the same species (see *Appendix 5—figure 1A*). In contrast, actively sensing target distance by observing the change in the target's azimuth during self-motion could partly resolve this ambiguity, provided the target is positioned at relative offset angle to the larva's heading. Moving straight toward a target will not produce any change in azimuth (see *Appendix 5—figure 1C*), while a given non-zero initial offset angle will change depending on the prey's distance and the travel distance of a motion bout.

Similar active-sensing techniques (*Egelhaaf et al., 2014*; *Ahissar and Assa, 2016*) are employed by insects that have been reported to move in particular angles to objects in order to generate optic flow and discern their proximity. Generally, off-axis approach strategies can extend beyond vision and may represent navigational trade-offs. For example, an off-axis strategy has been shown to improve object localization in echo-locating Egyptian fruit bats, and in these bats, it appears this strategy represents a strategical trade-off between positional accuracy and detection range (*Yovel et al., 2010*). In our case, the undershooting strategy may present a trade-off between minimizing the distance travelled to reach prey and the perceptual accuracy of prey distance.

This prey approach strategy predicts that bout-motion and distance perception operate in real-time closed-loop (*Ahissar and Assa, 2016*) and is consistent with evidence that larval bouts are not ballistic but utilize visual feedback during motion (*Portugues and Engert, 2011*; *Portugues et al., 2015*; *Jouary et al., 2016*), and with evidence that loss of visual feedback during approach can impair larvae from reaching capture distance (*Jouary et al., 2016*).

The retina could be directly computing the speed of moving edges (*Portugues et al., 2015*), while tuned specifically to prey-like stimuli by integrating retinal image motion with image size, which is in agreement with evidence that specific combinations of stimulus size and speed are required for hunting initiation (*Bianco and Engert, 2015*; *Barker and Baier, 2015*; *Semmelhack et al., 2014*; *Antinucci et al., 2019*). To generate this extra distance perceptual cue, which is based on image motion, would require that during forward bouts prey position is at an offset angle. From this, the prediction arises that prey located off-axis is more likely to be captured, which is consistent with the evidence from the distribution of prey-detection azimuths of *successful hunting episodes* (see *Figure 5—figure supplement 1*, which is bimodal although evidence from freely swimming larvae shows that hunting is initiated against prey in the frontal visual field just as well [*Bianco et al., 2011*; *Bolton et al., 2019*; *Bianco and Engert, 2015*; *Jouary et al., 2016*]). Curiously, other studies have observed similar bimodal distributions of prey-detection azimuth that are not conditioned on hunting outcome (*Bianco et al., 2011*; *Romano et al., 2015*; *Antinucci et al., 2019*), we are not sure what underlies these discrepancies between studies, but it may relate to whether the criteria used for detecting hunting initiation are solely based on eye-vergence or whether unilateral tail bends (J-turns) are also used.

## Learning the components of efficient behavior

Understanding the structure of learning is dependent on obtaining a valid description of how different hunting strategies can arise. We cannot exclude the possibility that larvae posses an adaptive toolbox (see *Todd and Gigerenzer, 2012*; *Todd and Gigerenzer, 2007*) of distinct, preset, hunting strategies. This would imply that the role of learning is to utilize experience to find the best match between the set of available hunting behaviors and the particular foraging environments a larva encounters. Alternatively, efficient hunting behavior could be the result of a learning process that progressively adapts the behavioral parameters of an innate hunting routine. In this case, adapted behavior may be manifested through incrementally modifying multiple independent behavioral parameters (see *Portugues and Engert, 2011*; *Ahrens et al., 2012*; *Severi et al., 2014*), or even a single behavioral parameter could suffice to explain multiple aspects of adapted behavior, if behavioral control turns-out to be low-dimensional. For example, modifying turn behavior could cause changes in the visual feedback obtained during prey pursuit (see previous section), and this could be causing changes in the kinematics of motion bouts, which have been shown to be affected by changes in visual input (*Portugues and Engert, 2011*; *Trivedi and Bollmann, 2013*; *Jouary et al., 2016*). In either case, the evidence here suggest that learning does not seem to modify individual behavioral components of the hunting sequence independently.

This conclusion is based on the observation that undershooting and fast long-range capture swims are strongly correlated most often in larvae raised with live prey (*Figure 7—figure*

*supplement 2*, *Figure 7—figure supplement 3*). This correlation is in not intrinsic, long-range capture swims are not an inevitable consequence of undershooting, because the opposite correlation (overshoot with fast-capture swims) also exists and is most often seen in prey-naive animals (NF,DF). The evidence also supports that the relationship between undershoot and capture speed is adjusted with experience (*Figure 7—figure supplement 2*), something we believe is an indirect consequence of a tendency to adjust capture speed with distance (*Figure 7—figure supplement 4*). Nevertheless, this relationship between capture speed and distance is also affected by experience (compare DF,LF against LF), something we posit relates to achieving suitable gape-timing for engulfing prey as larvae lunge forward during fast capture swims. We find that hunting strategies that combine fast capture swims executed from a longer distance are more likely to lead to consumption of prey (*Figure 7B*, *Figure 4*), and thus their ontogeny could be based on positive reinforcement through operant conditioning (*Skinner, 1938*). Undershooting on the other hand does not lead to an immediate reward, so how does experience reinforce and combine these two strategies?

Taking into account that undershooting could improve prey-distance perception during approach (see previous section) then the possibility arises that this behavior could be reinforced independently of hunting outcome, provided perception of prey during self-motion could be of intrinsically positive value (see *Kawashima et al., 2016*). Perhaps the simplest explanation is that correlations arise through the reinforcement of a shared gain control input, whose conditional reinforcement results in increasing all of these behavioral responses simultaneously. Alternatively, undershoot behavior could be independently reinforced by letting the conditioned stimulus that triggers the fast-capture swims also act as a reinforcing stimulus. This idea is based on higher-order learning processes, such as secondary conditioned reinforcement, or second-order conditioning, according to which stimuli paired with primary reinforcers acquire reinforcement properties themselves. A mechanism that allows for stimuli that predict rewards to become rewards themselves is believed to underlie the construction of adaptive and arbitrary long behavioral sequences (*Skinner, 1938*; *Williams, 1994*; *Enquist et al., 2016*).

However, previous attempts to establish classical or operant learning in larval zebrafish, by pairing visual cues to electric shock, failed to show learning at this early stage (*Valente et al., 2012*). A different, classical conditioning study showed enhanced tail response to the conditioned stimulus after pairing a tactile stimulus on the side of the body with a visual cue of a moving spot (*Aizenberg and Schuman, 2011*). Thus, although in theory a form of arbitrary behavioral chaining could be used to generate efficient behavior in a wide variety of experienced conditions (*Enquist et al., 2016*), it would appear here that learning is constrained to expecting specific information about the environment to instruct the parameters of particular behavior of the developing animal (*Bateson, 1981*; *Todd and Gigerenzer, 2007*). In this case, early foraging experience could allow larvae to learn and adapt their behavior to the food sources available in their environment. Although natural variability in learning speed is expected, our analysis on individual larvae suggests that the effect of a brief (two days) hunting experience is to rescue, or trigger the maturation of, initially ineffective hunting behavior. This is supported by the distribution of hunt efficiency (*Figure 3D*) in LF showing relatively lower numbers of low efficiency hunters compared to controls, which suggests that perhaps larvae with the lowest hunting efficiency are the ones who mostly benefit from early experience.

Finally, it is worth mentioning that the behavioral parameters of the control groups DF and NF are not always matching, and DF is also learning to adapt to its foraging conditions. For example DF has distinct capture speed and distance behavior to NF (*Figure 7—figure supplement 4*), while there is a small probability this may be counterproductive to its capture efficiency against live prey (*Figure 3B*).

## The neural basis of an adaptive hunting routine

Localizing the neural circuits that support experience-dependent changes to evoked hunt-rates (*Figure 2E*) draws attention towards the pretecto-hindbrain and the pretecto-hypothalamic pathways. This is because both of these have been associated with the release of hunting behavior in response to visual prey stimulation (*Muto et al., 2017*; *Semmelhack et al., 2014*; *Antinucci et al., 2019*; *Filosa et al., 2016*), and thus changes in their activity could be underlying the modulation of hunt-rates.

Yet, a companion study, which imaged the brains of partially restrained larvae (7 dpf) during hunt-initiation, did not find differences in tectal and pre-tectal activity between experienced and

naive larvae (*Oldfield et al., 2020*). Nevertheless, consistent with the higher evoked hunt-rates of experienced larvae, the functional connectivity between optic tectum and pre-tectum appears to increase, along with the probability that pretectal activation is followed by eye-vergence (*Oldfield et al., 2020*). The main difference observed was that forebrain activity (telencephalon and the habenula) in experienced larvae was higher during eye-vergence (hunting) compared to naive larvae. Further, ablation of the forebrain, in experienced larvae, demonstrated that although forebrain recruitment is not necessary for initiating hunting, it does contribute to the frequency of these events (*Oldfield et al., 2020*). Combined with evidence of increased functional connectivity between tectal periventricular neurons (PVN) and the forebrain's telencephalon in experienced larvae (*Oldfield et al., 2020*), it appears that experience-dependent adaptations to the hunt-rate are manifested via a tectal pathway that goes through the forebrain. This pathway appears to operate in parallel to those supporting innate hunt initiation, reflecting an organizational principle according to which pathways may be functionally classified into those supporting innate behavior and others supporting learned adaptations to it.

How can the forebrain modulate hunt-rates? The forebrain's diencephalon contains modulator neurons whose activity is correlated to multiple sensory modalities and to locomotor behavior (*Jay et al., 2015*; *Lambert et al., 2012*; *Reinig et al., 2017*). In terms of their effect on locomotion, evidence from ablation experiments suggest that diencephalic dopaminergic neuron recruitment could only explain changes in general swimming activity, and not finer locomotor behavior during prey approach (*Jay et al., 2015*), while detailed studies using tactile and visual stimuli suggest that dopaminergic action on swimming behavior maybe indirect, through sensory and proprioceptive processes (*Reinig et al., 2017*). In light of these prior studies, it would appear that the dopaminergic neurons of the diencephalon are not part of pathway that can stimulate hunt initiation. Perhaps a better candidate would be a forebrain pathway that targets the serotonergic system, which has been shown to be involved in modulating the release of hunting in response to visual prey recognition in a manner that depended on a larva's feeding-state (*Filosa et al., 2016*) or its internal foraging state (*Marques et al., 2020*). Identifying whether the activity of this system is also modified by experience could help to further unravel the neural processes that allow for experience to modify hunt-rates.

Similarly, experience dependent changes to turn-behavior (undershoot) may be supported by a pathway that runs parallel to the one known to control visual-evoked orienting turns towards prey. Visual-evoked orienting turns towards prey have attributed to the tectal-hindbrain pathway, which forms a sensory-map with a functionally segregated anatomy, of distinct descending retinotopically organized projections for approach and avoidance behavior, that can signal graded visual-evoked orienting turns towards prey (*Helmbrecht et al., 2018*; *Orger, 2016*; *Fajardo et al., 2013*; *Orger et al., 2008*; *Romano et al., 2015*). However, the development of this sensory-map is generally not driven by visual experience (see *Marachlian et al., 2018*; *Pietri et al., 2017*), and, in agreement with this, the tectal activity maps for prey location are not altered by experience (*Oldfield et al., 2020*). Given this evidence, changes in the tectal encoding of prey position do not underlie the differences in turn-behavior between prey-naive and experienced larvae observed in this study.

We posit that the modulation of turn-gain is most likely manifested downstream, perhaps at the level of reticulospinal neurons via the modulatory action of a pathway that runs parallel to the primary tectal-hindbrain one, such as the aforementioned diencephalic dopaminergic neurons (*Reinig et al., 2017*). However, further experiments would be required to establish if forebrain activity is associated to undershooting behavior in order to support this hypothesis.

Such turn-to-prey experiments are generally feasible and can be combined with functional imaging on partially restrained animals using virtual or real prey (*Bianco et al., 2011*; *Jouary et al., 2016*; *Oldfield et al., 2020*). By monitoring tail movement we may be able to infer the larva's intended turn magnitude (*Jouary et al., 2016*) with sufficient accuracy, and the relationship between turn-gain and forebrain activity. The mechanisms behind forebrain behavioral control could then be followed up by examining the potential role of descending dopaminergic action on turn-gain (see *Lambert et al., 2012*, but see *Reinig et al., 2017*). However, if forebrain circuits are involved in expressing both naive and experienced behavior incommensurably, then ablations may not be particularly revealing because behavioral aberrations would then be observed in both groups, naive and experienced. We posit that appropriately timed, optogenetic investigations of gain-of-function could

be more revealing on whether the forebrain controls the turn-to-prey gain during prey pursuit. We anticipate that the above pathways may also be involved in the kinematic adaptations of prey approach that result in increasing prey capture distance. Whole-brain imaging experiments combined with a closed-loop visual stimulation (*Jouary et al., 2016*) could be utilized to look for the brain structures that correlate with the changes in bout kinematics, such as bout duration (*Severi et al., 2014*), that drive changes in capture distance between naive and experienced larvae.

Although it has been possible to identify the learning centers of short-term motor adaptation (*Ahrens et al., 2012*), finding the circuits that support the longer term learning process presented here may prove more challenging. One reason is that contrasting brain activity between successful and failed capture episodes is not guaranteed to reveal the learning centers, because the required timescales of reward signaling are not yet known. Even if learning is steered by capture outcomes, and differences in brain activity following successful and failed capture manoeuvres reveal the circuits of learning, this would still require whole-brain imaging during unconstrained hunting behavior – something that is quite challenging technically.

Fortunately, the zebrafish community has already achieved such feats (*Cong et al., 2017*; *Kim et al., 2017*), with the first data of brain activity indicating a few seconds of sustained activity in hypothalamus, midbrain, and hindbrain following prey capture (*Cong et al., 2017*), while more recently, activation of hypothalamic dopaminergic neurons has been associated with capture success (*Marques et al., 2020*). However, imaging during unconstrained behavior is no panacea, because making causal inferences could become harder in comparison to more constrained experimental designs, which provide more control. For example, similar sustained hypothalamic dopaminergic activity has been previously observed in partially restrained animals following vigorous motor activity (*Reinig et al., 2017*), so fast-capture manoeuvres may be sufficient to explain some of the brain activity observed during unconstrained prey capture. Faced with behavioral variability, unknown process timescales and the lack of tightly controlled conditions it may prove difficult to ascertain how patterns of brain activity relate causally to adapted behavior.

We suggest that before attempting to identify the circuits of learning using imaging, it might be prudent to first behaviorally establish the reward signals and their timescales. We cannot rule out the possibility that exposure to prey may be sufficient for the ontogeny of improved hunting skills. In addition, relevant reward signals may not immediately follow capture success, but might be associated to food digestion or growth, and thus operate over longer time-scales. Future research focusing on important stimulus and timescale aspects in the ontogeny of efficient hunting behavior could shed light to the mechanisms of learning.

## Conclusion

In summary, we have demonstrated that prior experience of live prey modifies and associates components of the larval zebrafish hunting sequence in manner that improves their capture success. Our findings suggest that the ontogeny of hunting behavior relies on learning by experience to fully develop. Combined with prior attempts that failed to show conditioning in larval zebrafish (*Valente et al., 2012*), it appears that learning may be constrained to particular tasks at this early stage, but nevertheless it is sophisticated enough to alter a multidimensional behavioral parameter space so that innate goal-oriented behavior is improved. Such interactions between an innate behavior and learning are very common to the ontogeny of behavior (*Hinde, 1973*) and it is likely that the general learning principles, which allow experience to shape brain development, are conserved across species (*Warburton, 2003*). This study will pave the way for using zebrafish to study the neural circuits and mechanisms of learning that transform ethologically relevant experience into efficient natural behavior.

## Materials and methods

### Rearing

Fertilized embryos under natural spawning were collected at 10.30am from a mass embryo production system (MEPS), where they are developmental stage synchronized within 15-min collection intervals. These were visually inspected and 17 healthy embryos are selected and placed in each of three 9-cm Petri dishes, which were filled with 35 ml Daneau and labeled at random to define the rearing

group (NF,DF,LF). The embryo dishes were maintained in an incubator under the same conditions. of 28.5 in system water (pH 7.3, conductivity 550 µS) on a 14:10 hr light:dark cycle. On 1 dpf non-developing embryos were cleared (usually 0–3 dead ones), and from that day on the dish where cleared of debris and had half the water replaced on a daily basis.

Feeding initiated just prior to 5 dpf, around the time when larvae begin to initiate hunting, and continued until the beginning of 7 dpf with a single feed each day between the hours of 3 and 4 pm. Live-fed (LF) received ≈ 200 live Rotifers (*Brachionus plicatilis*) (sized between 0.3 mm and 0.05), usually in 1–3 ml of 2 ppm water, the volume depended on the density of the culture on the day. The Non-Fed group (NF) received 2 ppt salt water, to control for treatment and salinity, in a volume matched to one supplied to the Rotifer fed dish (LF, 1–3 ml) on that day. The health or survival of the NF group was not impacted up to 7 dpf during which time their yolk-sack energy store appears sufficient to keep them healthy, and this is in agreement with studies showing that survival rates remain unaffected even if feeding commences on 8 dpf (*Hernandez et al., 2018*). The Dry-fed (DF) group receives grounded growth food suspended in the same amount of 2 ppm salt water as the one delivered to other groups. The DF food is grounded with mortar and pestle (Sera Micron or Ketting Gemma 75), then suspended in 2 ppm water and centrifuged in 800 rpm for 20 s. The suspension, which mostly contains particles smaller than Rrotifer typical size, is used for feeding such that the visual experience of moving dots is minimized for the DF group but nevertheless they receive a nutrition.

## Behavioral recording

At 7dpf individual larvae from each group were transferred to 35 mm petridishes (5 ml Daneau) and were allowed to acclimatize for 40 min-2 hr in the test conditions prior to each video recording. The recording protocol's timeline is shown on *Figure 1*. Each larvae was recorded in two test conditions. First settled and recorded in a prey-free (empty) arena and then, following the addition of ≈ 30 live Rotifers, larvae are left to settle prior to being recorded with live prey conditions. For the empty test conditions, individual larvae where randomly picked from the 9 cm group rearing dish, washed by transferring to a clear Daneau bath, and then transferred to their individual 35 mm dish containing 5 ml of filtered Daneau water, so that free-floating impurities that could trigger hunt events are minimized. For the live-prey test conditions, we added ≈ 30 live rotifers to the same dish as above, and then topped up prey numbers prior to placing the dish on the recording rig. Once the dish was transferred onto the recoding rig, we let it settle for 5–10 min before starting the recording software.

Our recording setup is dark-field illuminated via a custom made light-ring composed of 7 infrared (835 nm) emitting diodes (VSMY98545) that helped provide high-contrast images of both small prey particles and larval features. These are arranged appropriately such that illumination is uniformly distributed by the converging IR light beams onto the area of the circular 35 mm petridish, therefore eliminating light fluctuations and canceling any directional preferences arising from NIR light sources (*Hartmann et al., 2018*). We provided a total of ≈250 mW to the light-ring and tried to keep power low to avoid any thermal currents in the water. Below the arena sits a Chameleon 3 FLIR camera, with 50 mm/F2.8 lens kit MVL50M23, supplemented by a 5 mm lens spacer (CML05) to provide x3.5 magnification, and an long-pass filter so as to record in IR only. Above the arena sits a frosted-glass on which visible light is diffusely reflected from a directed lab-bench light source, set-up on either side of the rig (see Appendix 2). This provided sufficient lighting for the larvae to be able to see and track prey.

Video recording was controlled via custom recording software that allowed us to limit the total recording time to 10 min and to minimize video data, by not recording when the larva is not within a central region of interest (ROI). This was set to a 25 mm diameter circle in the center of the 35 mm circular petridish. Behavior was recorded at 410 images per second with a resolution 640 × 512. Raw image sequences were then converted into compressed video files using the highly efficiency H.264 compression codec. Recording events were automatically triggered when an object of sufficient size (> 120px area) entered the central ROI, thus ignoring behavior near the edges of the arena. Each triggered recording event was set to have a minimum duration of 30 s., and an initial recording event is automatically triggered at the start of each experiment, even if the larva is not within the ROI for this event. This initial short clip allows us to automatically estimate and verify the number of prey at the start of each experiment. If a larva was still within the ROI at expiration of the

10 timeout period, the recording time was automatically extended up to a maximum of 2 min to wait for the larva to exit the ROI. This aimed to avoid the 10 min timeout interrupting an ongoing hunting episode. If a larvae triggered no recording events in the empty conditions it was then rejected and replaced by a new one on which the recording protocol was restarted, beginning with settling in empty conditions.

Recording began at 11 am and continued throughout the day usually ending around 10pm. During the day, room temperature varied from 21°C to 26°C. Recordings from all groups were being balanced for time of day, to control for circadian and temperature effects, as well as batch variability, by recording the same number of larvae per group on each recording day. In order to determine the appropriate sample size, we performed a standard power calculation requiring an effect size on hunting efficiency of 0.5 (z-score), significance level of 5% and power 80%, resulting in approximately 60 animals per group based on a two-sample t-test. Overall, our dataset includes 15 batches, in total 180 larva (60 for each rearing group), which were obtained between 16 Nov 2017 and 21 August 2018. Behavioral tracking was conducted offline and was extracted from collected videos using custom video analysis software.

## Behavioral tracking

### Larva body tracking

Custom tracking software was written in C++, using QT for GUI development and OpenCV's image processing routines (*Bradski, 2000*). A background model is computed by employing OpenCV's mixture of Gaussians (MOG) background model on the initial 100 frames of video with a learning rate of 1/400, which is then set to a nominal rate of 1/1000. On each video frame, after extracting foreground objects, we filtered for blob area to identify a rectangular frame region that contains the larva. Because larva heads are very similar, we found that the orientation and position can then be easily and quickly located within this subregion using template matching. We compiled a small (20) library of larva head samples sized $22 \times 33$ px and replicated each sample across 360 rotations with a resolution of 1°. We then utilized these in our tracking software to do template matching and identify the position and rotation of the larva's head. The identified template rectangle framed the eyes and swim bladder, and its center, which was located near the anterior end of the swim bladder, was used as the reference point for tracking the larva's position.

### Eye tracking

We isolate head segment as matched by template. We upsample the head image, doubling image width and height, then we obtain a mean intensity value by sampling points along a elliptic arc passing through both eyes. We obtain three threshold values after ranking the sampled intensities the median, the 60 and the 85 percentile values, which are then used to to threshold the head image to segment the eyes at different intensities. The edges from each of the thresholded images is then extracted using Laplace edge detection. These are then combined and passed on to the ellipse detection algorithm. We isolate the detected edges from each eye separately by splitting the edges images into left and right panes. These are then independently processed by our customized implementation of a fast ellipsoid detection algorithm (*Xie and Ji, 2002*). The ellipsoids are scored based on the number of edge pixels they overlap with, and the highest scoring ellipsoids are considered to provide the best fit for eye shape. The angle of each eye is then read out as the angle of each eye's major axis relative to the body orientation. Noise from each of eye-angle trajectories is then suppressed during the data processing stage by passing the eye-data through a 4th order Butterworth low-pass digital filter (cut-off $\approx 28$ Hz) Eye vergence is computed as left eye - right eye angle, letting clock-wise being positive angles.

### Tail motion

The tail motion is tracked by fitting a spine of 8 points that approximates tail length and curvature. We employ two methods for fitting the tail spine. The first is employed on every video frame and it uses pixel intensity to adjust spine points to detect tail midline by utilizing the fact that the tail appears brighter than background in our images. The image of the larva is Gaussian blurred, and the algorithm adjusts existing spine segment angles to track the center of mass of image intensity. Each spine segment is taken in order, starting from the most anterior-proximal spine segment, and

then pixel intensity $i$ is sampled along the arc of drawn from rotating the spine-segment across angles within $\theta = -40 \cdots 40$ from its current position. Center of mass is calculated in vector containing 80 intensity sample points $COM_i = \sum_{r=0}^{80} r * i_r / \sum_r i_r$ The segments orientation is fixed to the angle with the maximum sampled intensity. The second method is employed more irregularly (every four frames), and it aims to position the spine within the limits of the body contour but additionally to fix spine length appropriately. This method uses a variational method to best position the eight point spine within a simplified, smoothed, larval contour shape made up of 90 points. By varying the spine parameters of length and angle, a Jacobian of matrix is computed and then a gradient in parameter space is computed. Gradient descent minimizes a cost function defined to be the sum of the distances of each spine point from the closest contour edge with an additional cost for fitting short tail lengths, therefore favoring the fitting of longer spines that are contained within the contour. The extracted angles of each spine segment is then noise filtered, during the data processing stage, through a Butterworth fourth-order band-pass filter (4–123 Hz).

## Detecting hunt events and labelling

Hunt events were detected via eye-vergence (*Bianco et al., 2011*). We define the start of hunting episode as the time when eyes verge beyond 45° with each eye being at least 19° inwards, and the end as the time when they diverge back out of the above range. Eye vergence needs to last at least 100 video frames (recording at 410 fps), and hunt-events need to be at least 300 frames apart, otherwise they are concatenated. The isolated video frames from the hunt events detected in evoked (prey) test conditions were played back to an independent observer, who was blind to the rearing group. They were allowed to observe the hunt-event for as long as they needed and then they were given a choice of labels to assign to the outcome of the event that included: *Capture success with strike*, *Capture success no strike*, *Capture failure no strike*, *Capture failure with strike*, *Failure no strike* (larva reached near the target and aborted or failed), *No target* (indicating events where no prey could be seen to be tracked).

## Bout detection and first turn to prey

Our data analysis was conducted via custom scripts in R (*R Development Core Team, 2019*). We identify the start of hunting episode as the time when eyes verge beyond 45° with each eye being at least 19° inwards. Larval speed is measured by tracking the center of the detected head template position (see above), and smoothed using a low pass Butterworth filter (24 Hz). The capture bout is the last bout prior to the head of the larva passing from the position of the prey, and we exclude any bout that sometimes can occur immediately following prey capture (*Marques et al., 2018*). In the case that hunting mode is initiated in conditions where there are multiple prey in the direction the larva faces, we consider the hunt initiation to be prior to the turn that unambiguously identifies the prey item that is being tracked and which the larva will attempt to capture.

## Measuring larva size and prey distance

The distances are calculated based using an estimate of mm per pixel in the video calculated by measuring the diameter of the 35 mm dish on screen in pixels. This mm/pixel ratio was estimated to be 35/790, that is, approximately 44 µm per pixel, and this scaling was used for all measurements of length and distance from images. For reference, single pixel errors translate to an error distance of approximately 0.05 mm.

Larval body lengths were measured on snapshots of larvae in straight posture by taking a straight line from the edge of the mouth-point to the point where the tail point vanishes (*Appendix 1—figure 1A*). Distances to prey were measured at the onset of the capture bout from the tip of the mouth point, while larval capture speed was tracked via a point on the head located near the anterior end of the swim bladder. Given our image resolution, single pixel errors are ≈ 10% when it comes to measuring distance from prey. Nevertheless, our measured distance from prey at the time of the capture strike are comparable to previously reported ranges (*Marques et al., 2018*).

## Bayesian statistics and notation

For a given dataset *D* described by a model *M*, Bayesian inference allows us to quantify the posterior distributions of model parameters (θ) according to the Bayes' theorem:

$$P(\theta|D,M) = \frac{P(D|\theta,M) \cdot P(\theta|M)}{P(D|M)}$$

where $P(D|\theta,M)$ and $P(\theta|M)$ correspond respectively to the data likelihood and the prior distribution of the model parameters. The denominator $P(D|\theta,M)$, also known as model evidence, defines the normalization factor of the posterior distribution.

We employed a probabilistic programming R package RJags (*Plummer, 2019*) which allows us to estimate posterior distributions using the Markov Chain Monte Carlo method (MCMC). With this approach we can generate random samples distributed according to the posterior distribution of the model parameters and use them to calculate averages and uncertainties of relevant variables. In the next sections, when modeling prior distribution and data likelihood we used the following parametrization of standard probability distributions:

- Normal distribution: $\mathcal{N}(\mu, t)$, with mean µ and precision $t = 1/\sigma^2$ defined as the inverse of the variance.
- Negative binomial distribution: $\mathrm{NB}(p, r)$, with $0 < p \leq 1$ being the probability, and $r$ being the size parameter ($r \geq 0$)
- Gamma and inverse gamma distributions: $\mathrm{Gamma}(a, b)$; $\mathrm{InvGamma}(a, b)$, with shape $a$ and rate $b$.

Finally, in the text and figure captions we evaluate probabilities of the form $P[a<b]$, where we compare posteriors between two conditions by taking a large number $n$ samples from each MCMC and count the number of samples where $a>b$, and then normalize by $n$ to get an estimate of P.

## Statistical modeling of hunt rates

We counted the number of detected hunt-episodes recorded for each larvae in spontaneous and evoked test conditions (as described in Materials and methods) and then used Bayesian inference to estimate the mean hunt-event rate for each group. Here, a single Poisson is not sufficient to model the occurrence of hunt events in a group of larvae, but instead a model that assumes a mixture of Poisson processes of various rates $\lambda_i$ is required to characterize hunt-activity in the group's population. This mixture of rates aims to account for natural behavioral variability in hunt-rates, expected even among larvae of the same rearing group and test conditions. For this, we employed a Gamma-Poisson mixture, where the number of hunt events $h$ conditional to hunt rate $\lambda$ in each recording time period (fixed to 10 min) is distributed as a Poisson distribution

$$p(h|\lambda) = \frac{\lambda^h exp(-\lambda)}{h!},$$

while hunt rates within the population are Gamma distributed

$$p(\lambda; \alpha, \beta) = \frac{\beta^\alpha}{\Gamma(\alpha)} \lambda^{\alpha-1} \exp(-\beta\lambda).$$

with hyperparameters α and β.

With this setting, the marginal distribution on the number of hunt events is a negative binomial $\mathrm{NB}(r = \alpha, p = \beta(1+\beta)^{-1})$, which provides a simple model with two parameters to characterize and regress each group's mixture of hunt rates. In particular, we can express the mean hunt rate as

$$\lambda = \frac{\alpha}{\beta} = \frac{r(1-p)}{p}$$

Once we infer appropriate parameters of $\mathrm{NB}(r, p)$ that fit the hunt rate data $h$ of each group, we can then use those parameters to extract a distribution for the mean hunt-rate. Specifically, the model we used was $h = \mathrm{NB}(r, q)$ with priors for $q = \mathcal{U}(0, 1)$ and $r = \mathrm{Gamma}(1, 1)$, while the $\mathrm{Gamma}(\alpha, \beta)$ distribution's parameters, shape and rate respectively, can be recovered as $\beta = q/(1-q)$ and $\alpha = r$.

The model code is available on GitHub (*Lagogiannis, 2020*; copy archived at https://github.com/elifesciences-publications/ontogenyofhunting_pub).

## Statistical modelling of hunt duration

We model the total hunt frames that a larva spends hunting during the recording interval. We take the total number of frames in a video in which a larvae was engaged in hunting-mode to be a stochastic quantity composed of individual hunt-frame events that occur with a rate $\lambda_f$. By counting the number of hunt-frames, the model of hunt-duration becomes equivalent to the hunt frequency model above, which counts the number of hunt-mode onsets for each larvae, and thus the negative binomial can be used here as well. Instead of counting hunt-events, we here count the total number of video frames each larvae spent in hunting mode. Note, we excluded larvae that produced no hunt events, and thus had a total hunt duration of zero frames. This is because hunt events are defined to have a minimum frame duration of 100 frames, and larvae without any detected hunt events make the hunt-duration distribution discontinuous.Using a model very similar to the one used for hunt-rates (Materials and methods) we estimated the density of mean hunt-duration through the negative binomial $\mu = R_{\mathrm{fps}}r(1-q)/(q)$, where $R_{\mathrm{fps}}$ is the fps of video acquisition (410 fps), while $q$ and $r$ are the inferred model parameters from the duration data.

## Statistical modeling of hunt efficiency

We define the capture efficiency of each larva as the fraction of hunt events it performed that ended with successful capture, over the total number of hunt events in which prey capture was attempted. Larvae which did not perform any capture attempts are excluded from this analysis.

We utilized the earlier hunt-rate model (Materials and methods), to model the distribution of capture attempts $h$ of each larval group,

$$h \sim \mathrm{NB}(r,q) \tag{1}$$
$$q \sim \mathcal{U}(l,u) \quad l=0,\, u=1 \tag{2}$$
$$r \sim \mathrm{Gamma}(a,b) \quad a=1,\, b=1 \tag{3}$$

here it is extended to infer the probability $p_s$ of successfully capturing a prey. For this, we used a binomial distribution to model the number $N_s$ of successful captures in the 10 min recording period

$$N_s \sim \mathrm{B}(p_s,h) \tag{5}$$
$$p_s \sim \mathrm{Beta}(\alpha,\beta) \quad \alpha=1,\, \beta=1. \tag{6}$$

which depends on the total number of events $h$. By using Bayesian inference, we estimated the distribution of capture probability $p_s$ for each group independently.

## Linear regression of turn to prey behavior

The data used for this model are the magnitude of the first turn toward prey $\phi$ and the prey azimuth $\theta$, prior to the larva turning toward the prey. Data points are pooled for each rearing group. We characterized each group independently using a linear model and performed a Bayesian analysis of the linear coefficients, which enables us to compare rearing groups using their posterior distributions. The linear regression model for each group is defined as:

$$\phi \sim \mathcal{N}(\beta_0 + \beta_1\theta, 1/\sigma^2), \tag{7}$$

which assumes the errors are independent and identically distributed as normal random variables with mean zero and variance $\sigma^2$, for which we used an inverse gamma prior

$$\sigma^2 \sim \mathrm{InvGamma}(s,r), \quad s=5,\, r=2 \tag{8}$$

while for the linear coefficients we used

$$\beta_0 \sim \mathcal{N}(\mu,t) \quad \mu=0,\, t=2 \tag{9}$$
$$\beta_1 \sim \mathcal{N}(\mu,t_1) \quad \mu=0 \tag{10}$$
$$t_1 \sim \mathrm{Gamma}(a,b) \quad a=1,\, b=1 \tag{11}$$

Our aim is to update the initial guess distributions for the model's $\beta_0, \beta_1, \sigma^2$ for each rearing group according to its relevant data points. We then report and compare the density estimates for

the slope parameter $\beta_1$ of each group's model, after smoothing using a Gaussian kernel (BW = 0.01), on *Figure 5E*.

## Classification of fast and slow capture swims

The final motion with which a prey is captured is executed with a range of speeds indicating difference in vigour. Broadly speaking, we observe two types of capture motions, a stereotyped fast speed capture swim, which is usually successful executed with the larva standing at some distance to the prey, and a weak bite capture, which can only be successful if executed when the prey is very close or touching the larva.

Our tracking system is not able to automatically give us reliable and accurate information on the distance to prey prior to capture, as small position errors get magnified due to low video resolution. Thus, we decided to conduct supervised tracking of the specific hunt events, to minimize errors but also verify the validity of our data. The video frames during final capture were re-analyzed and the distance from the edge of the mouth-point $d$ to the prey position was measured with the help of a user distance measurement tool that we built into our tracker (see Materials and methods). These distances were then used as data on distance to the prey at prior to the capture bout. Capture speed $s$ is measured as the peak speed of the final bout occurring between the frames starting from the frame of capture bout onset (on which the prey-distance is manually measured) and up to the first time the larva's speed goes below a motion detection threshold (4 mm/sec) in the frames proceeding the time of when the larva's centroid has reached the closest point to the prey.

Capture speed $s^c$ and distance to prey $d^c$ were modeled using a mixture of two bi-variate normal distributions to accommodate slow and fast events labeled using the index $c = \{s, f\}$:

$$\begin{pmatrix} s^c \\ d^c \end{pmatrix} \sim \mathcal{N}\left[ \begin{pmatrix} \mu_s^c \\ \mu_d^c \end{pmatrix}, \begin{pmatrix} \sigma_s^{c2} & \rho^c \sigma_s^c \sigma_d \\ \rho^c \sigma_s^c \sigma_d & \sigma_d^2 \end{pmatrix} \right], \tag{12}$$

where $\mu_s^c$ and $\mu_d^c$ are respectively the mean speed and distance to prey, $\sigma_s^c$ and $\sigma_d^c$ the standard deviations and $\rho^c$ is the correlation coefficient between speed and distance for each cluster $c$. We used the following priors on the mean and covariance parameters

$$\mu_s^s \sim \mathcal{N}(\mu, t) \quad \mu = 5, \, t = 10^{\frac{1}{2}} \tag{13}$$
$$\mu_d^s \sim \mathcal{N}(\mu, t) \quad \mu = 1/2, \, t = 10 \tag{14}$$
$$\sigma_s^s \sim \mathcal{U}(a, b) \quad a = 0, \, b = 2 \tag{15}$$
$$\sigma^d \sim \mathcal{U}(a, b) \quad a = 0, \, b = 1 \tag{16}$$
$$\rho^s \sim \mathcal{U}(a, b) \quad a = -1, \, b = 1 \tag{17}$$

for the slow group and

$$\mu_s^f \sim \mathcal{N}(\mu, t) \quad \mu = 35, \, t = 10^{\frac{1}{2}} \tag{18}$$
$$\mu_d^f \sim \mathcal{N}(\mu, t) \quad \mu = 1/2, \, t = 10 \tag{19}$$
$$\sigma_s^f \sim \mathcal{U}(a, b) \quad a = 0, \, b = 10 \tag{20}$$
$$\sigma_d \sim \mathcal{U}(a, b) \quad a = 0, \, b = 1, \tag{21}$$
$$\rho^f \sim \mathcal{U}(a, b) \quad a = -1, \, b = 1 \tag{22}$$

for the fast group, with a narrow $\sigma^s$ to add the prior belief that low capture speeds are positioned in narrow range at the lower end in the range of capture speeds observed. The prior on group membership was 0.5 for fast and slow capture.

The standard deviation for capture speed is wider and is representing the prior belief that fast capture speeds occupy a wide range of speeds above the slow captures modes. Indeed beyond slow captures, capture speeds do not appear stereotyped and are seen to vary probably in relation to other parameters such as distance to prey. Note the prior for distance to prey is set identically for both clusters and as such we expect the data to inform the mean distance of each cluster.

The probability of membership on either cluster $c \in 0, 1$ for fast/slow is estimated from a normal distribution with as $p_f = \mathcal{N}(\sum I(c_i = 1)/N, 0.03)$, where $I(c_i = 1)$ is equal to 1 if data point $i$ has been

assigned to the fast cluster. Data points were assigned to the fast cluster if the expected cluster membership label was $E[c]>0.7$, otherwise they were considered as slow capture swims.

## Statistical modelling of group behavior

We built a hierarchical statistical model to estimate mean group behavior that is based on model estimates of mean hunting behavior per larva. This is a generalization of our earlier model of capture-speed and distance, only here the top level structure is single multivariate normal distribution that models $X^g$, a vector containing estimates of mean capture speed $S$, distance to prey $D$ and turn-ratio $T$ for each larva. Details of this model and example code can be found in the code repository https://github.com/kostasl/ontogenyofhunting_pub/blob/master/stat_3DLarvaGroupBehaviour.r (*Lagogiannis, 2020*).

## PCA of larval hunting behavior and predicting efficiency

Principal component analysis can reduce the dimensions needed to describe a large set of correlated predictor variables to a smaller, less correlated set of covariates, that nevertheless maintains most of the information in the larger set. A subset of the resulting covariate components can then be used to regress an outcome variable, effectively producing a model that predicts a response based on a subset of principal components.

The process of obtaining principal components involves constructing a covariance matrix $\mathbf{A}$ of our observation data and then calculating its eigen decomposition. For our analysis, we constructed a matrix of vectors, each one representing the hunt behavior of each larvae estimated from measurements taken from successful hunt episodes alone. For each larva $i$ we defined vector $\vec{v}_i = \langle e, n, E(c), E(d), E(\gamma)\rangle$, where $n = N_s + N_f$ is the capture attempts recorded for larva $i$, $e = N_s/n$ denotes its capture efficiency, $E(d)$ is the mean distance to prey estimated from the onset of the capture bout across a larva's hunt events, and $E(\gamma)$ is mean turn-ratio at the initial turn towards prey. For *Figure 7*, we derive the expected values per larva from the results of our earlier group model (see Materials and methods), while in *Figure 7—figure supplement 5* we used empirical means ($E(x) = \sum_n x_i/n$) to estimate the mean hunt behavior. In order to ignore any scale effects of covariance on PCA, we standardized the variables using

$$\dot{x}_i = x_i - \mu_x/\sigma_x. \tag{23}$$

With the exception of turn-ratio where $\mu_x = 1$, $\mu_x$ was set to the mean value of each behavioral variable $x$ calculated across larval vectors.

Each principal component packs correlated variables that could possibly act as better predictors and provide compact regression models in situations where there many predictor variables and relatively few samples. We used the *pls* package (*Wehrens and Mevik, 2007*) in R (*R Development Core Team, 2019*), to conduct principal component regression of larval efficiency using a linear model and the principal components of matrix of hunt behavior estimators of $\vec{v}_i$. The algorithm reported that using three PCs gave the smallest root mean square prediction error with coefficient of variation CV = 0.133, capturing 31.7% of efficiency variance.

## Acknowledgements

We thank Sabine Issop for help with manual labeling of hunt events, the fish facility staff at King's College London for their excellent fish husbandry, Elina Jacobs and Adil Khan of King's College London for their feedback on our manuscript, and KL would also like to thank Andrew Bolton, of Harvard University, for the feedback on manuscript and the cross-validation of observations, Sarah Stednitz, of Max-Planck Institute for Biological Cybernetics, for her critical feedback on the relationship between larval size and hunting ability.

## Additional information

### Funding

| Funder | Grant reference number | Author |
| --- | --- | --- |
| Wellcome | 204788/Z/16/Z | Konstantinos Lagogiannis<br>Giovanni Diana<br>Martin P Meyer |

The funders had no role in study design, data collection and interpretation, or the decision to submit the work for publication.

### Author contributions

Konstantinos Lagogiannis, Conceptualization, Data curation, Software, Formal analysis, Investigation, Visualization, Methodology, Writing - original draft, Project administration, Writing - review and editing; Giovanni Diana, Formal analysis, Visualization, Methodology, Writing - review and editing; Martin P Meyer, Conceptualization, Resources, Supervision, Funding acquisition, Investigation, Methodology, Writing - original draft, Project administration, Writing - review and editing

### Author ORCIDs

Konstantinos Lagogiannis (iD) https://orcid.org/0000-0001-9349-801X
Giovanni Diana (iD) http://orcid.org/0000-0001-7497-5271
Martin P Meyer (iD) https://orcid.org/0000-0001-8337-630X

### Ethics

Animal experimentation: This work was approved by the local Animal Care and Use Committee (King's College London) and was performed in accordance with the Animals (Scientific Procedures) Act, 1986, under license from the United Kingdom Home Office Licence number P9090AEFD. All primary data included in the manuscript came from the use of zebrafish larvae. All procedures were non-invasive and classified as mild according to the Animals Act 1986 and as defined by the United Kingdom Home Office, in order to minimize animal suffering. At the end of regulated procedures, animals were culled using a schedule 1 method (terminal dose of MS222).

### Decision letter and Author response

Decision letter https://doi.org/10.7554/eLife.55119.sa1
Author response https://doi.org/10.7554/eLife.55119.sa2

## Additional files

### Supplementary files

• Transparent reporting form

### Data availability

All data analyzed during this study are included in the supporting github repository.

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

# Appendix 1

## Feeding regimes have detectable effects on growth

To test whether the different feeding regimes influence the development and growth of larvae we measured body lengths, as shown on *Figure 1*, for each feeding group from 7dpf images of larvae taken from the recording videos ($n = 37$ NF, $n = 40$ LF and $n = 57$ DF). *Figure 2K–M* show distribution of measured body lengths in the different feeding regimes, while *Figure 2N* estimates the mean larva lengths of each feeding group using a Gaussian model, and we find these are very close between groups. The probability density for each feeding group (*Figure 2K*) show small but distinct differences in mean body length, with live-fed larvae showing a statistical significant different mean, 0.2 mm longer on average from the not-fed group. These results also reveal that the dry-fed group does indeed receive some nutritional value from the feeding regime we provided, and thus can act as control for the effects of nutrition.

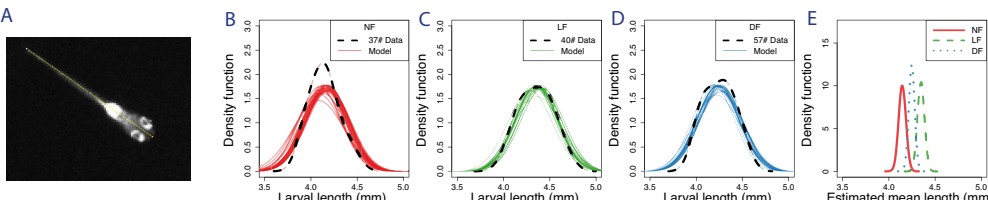

**Appendix 1—figure 1.** Analysis of larval length shows small statistical differences in mean length, with larvae on live-prey being the longest. (**A**) Larval length is measured in pixels from mouth point to edge of tail on video frames when the larva is not in hunting mode and in straight posture. As the tail-fin is not visible or measured, our measurements are equivalent to the established developmental measures of standard-length (SL) (*Parichy et al., 2009*), which we convert from pixels to mm using an estimate of the mm/px calibration of our setup. (**B–D**) Distribution of body lengths in the different feeding regimes. Dotted black lines indicate kernel density smoothed distributions of measured larva lengths (Gaussian kernel BW = 0.1) and solid lines show 30 samples of likely body length distributions based on a Gaussian model fit, whose parameters (μ,σ) were estimated from the data using Bayesian inference. (**E**) The estimated probability densities of mean body length based on Gaussian model fitting. The estimated mean SL of each feeding group (NF = 4.15 mm, LF = 4.37 mm, DF = 4.21 mm) are distinct, ($P[LF>NF]>0.99$), ($P[LF>DF]>0.98$) and ($P[DF>NF]>0.97$), with LF having the largest mean length, being larger to NF by $\mu_{LF-NF} \approx 0.2$ mm and $\mu_{LF-DF} \approx 0.1$ mm.

The relationship of larval length against hunting ability is examined in n *Figure 3—figure supplement 1*, showing that hunting ability is not correlated to larval size in controls, but only in live-food reared fish. This suggests that hunting ability can explain larval growth while the differences in larval size found within each group do not affect prey capture ability, at this stage in development. Interestingly we find evidence that larger DF larvae are likely worse hunters, suggesting that the skills for feeding on dry food are counter productive against live-prey.

# Appendix 2

## Recording apparatus

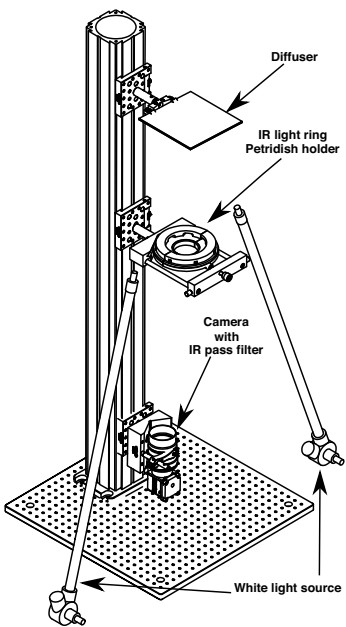

**Appendix 2—figure 1.** Behavior imaging system. Behavior is recorded from below a 35 mm petridish arena, with the camera connected to custom software such that after triggering a default recording event at the start experiment, subsequent recordings events are triggered when the larva is within a circular region of interest (ROI). An a long-pass filter is fitted on the camera lens, and a custom IR light-ring illuminates the arena uniformly. Diffused light is used to illuminate the arena, that obtained via lab-bench light-source pointing upwards onto a frosted glass that sits above the arena. The event ROI is set such that Behavior is recorded only when larval is sufficiently away from the edge of the petridish. The recording session timeout is 10 min, beyond which time new recording events are not triggered. The maximum duration of recording events is limited to two mins. If the larva is still in the ROI after those 2 min, and the maximum experimental period (10 min) timeout has not been exceeded, then a new recording event is initiated.

## Appendix 3

### Hunting sequence tracker frames

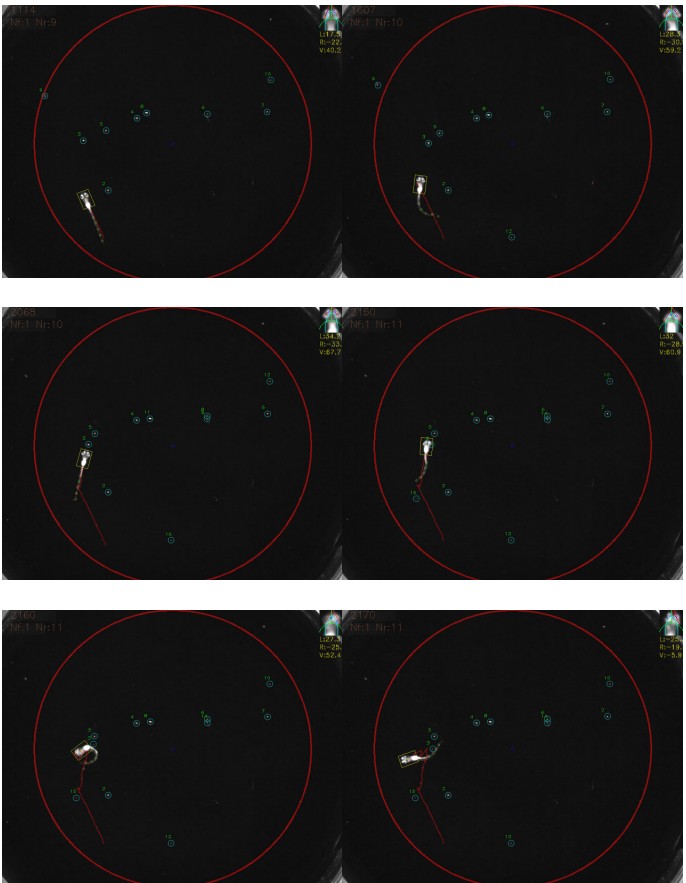

**Appendix 3—figure 1.** Example frames from a hunting sequence being tracked via our software showing initial detection of prey, turn to prey, approach and capture. The eye vergence angle is detected and shown at the top right of the screen.

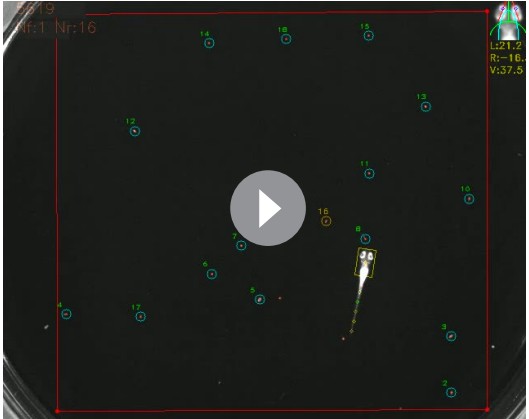

**Appendix 3—video 1.** Re-tracking a successful capture event within a custom ROI: slow playback speed x1/16.
https://elifesciences.org/articles/55119#A3video1

## Appendix 4

### Capture travel-time to prey against capture distance

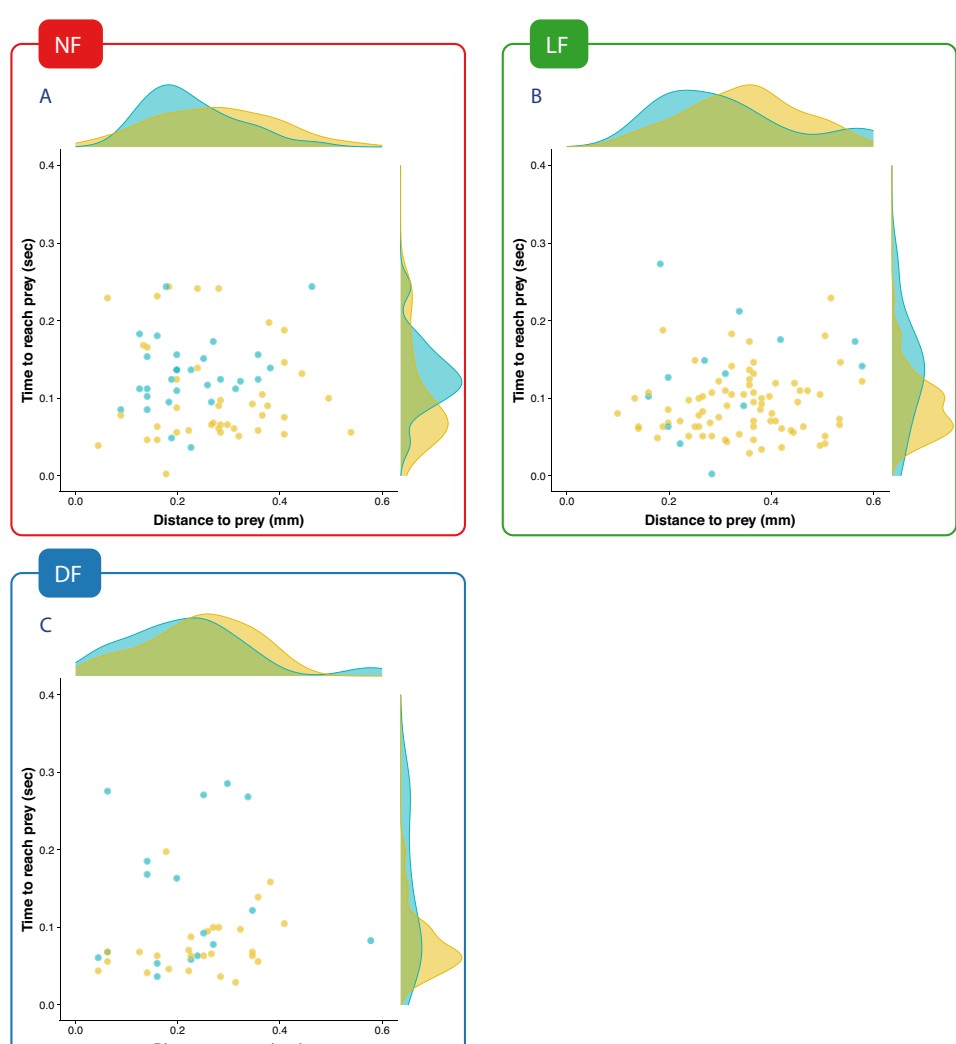

**Appendix 4—figure 1.** The time it takes to reach prey for successful fast capture swims does not strongly depend on the distance from which these are executed. Points are coloured according to the classification in capture speed as in *Figure 4* Yellow: fast-capture swims, cyan:slow capture swims. In all groups the time to reach prey is longer for the capture swims that were clustered as slow, while for fast-captures swims the timing is more compactly distributed below 0.15 s. Maintaining such timing would require adjusting capture speed with prey-distance. It appears that LF (**B**) can regularly do this even for prey distances beyond 0.4 mm, where successful hunt episodes from NF, DF (**A,C**) are rare. The overall distributions of time-to-reach prey match the ranges of maximum gape-timing reported in *Mearns et al., 2020*, and therefore support the hypothesis that time of maximum gape is synchronized to the time the larva reaches/hits prey during a capture swim.

## Appendix 5

### Calculating the sensitivity of prey azimuth $\theta$ to distance $r$

We would like to analyse how the relative prey motion can affect the aiming accuracy of larvae, and how this is influenced by the prey speed and its relative position.

*Figure 1* depicts how prey azimuth θ is modified differently depending on whether a prey is originally located at distance $\alpha_1$ (near) or $\alpha_2$ (far), before it travels a distance $\beta$ orthogonally to the larva's midline. It is easy to see that the prey that was originally near at $\alpha_1$ is the one that causes a larger change in prey-azimuth $\delta\theta$ after it travels distance $\beta$. Therefore, assuming prey travels at a speed $\Delta\beta/\Delta t$, the azimuth's angular velocity $\theta'$, will also dependent on prey distance. This movement could significantly affect aiming accuracy, and therefore capture success, depending on whether prey speed is sufficient to cause large deviations in $\theta$ during movement over the last inter-bout interval preceding the capture swim.

To examine this, we let $\alpha,\beta$ be Cartesian coordinates of prey relative to the position of a larva's mouth, with $\alpha$ denote the straight line distance pointing along the larva's heading extending from the midline, and $\beta$ being the prey distance normal to this midline axis (see *Appendix 5—figure 1*). We define a function for prey azimuth $\theta = f(\beta, r)$ (the bearing angle to prey), with $r$ being the straight line distance to prey, and proceed to obtain the total derivative, which we can use to make make a linear approximation of change of θ about prey position (α,β).

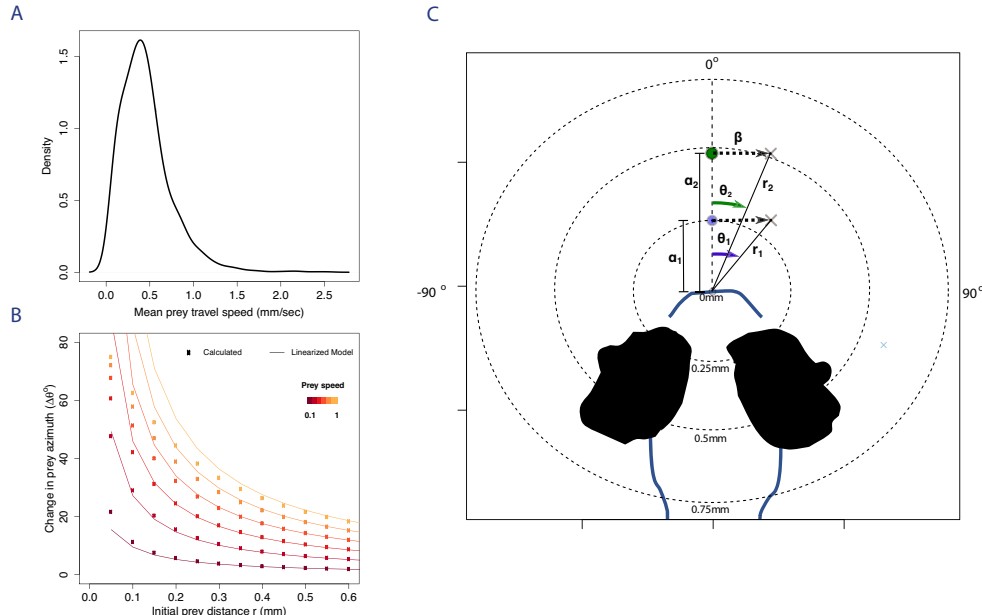

**Appendix 5—figure 1.** The aiming accuracy prior to a capture swim is strongly dependent on capture distance for the observed prey speeds. (**A**) The distribution of mean prey travel speeds is estimated by measuring the total displacements per second from a random sample of videos each containing approximately $n \approx 15 - 30$ freely swimming Rotifers. By analysing tracks of prey motion we estimated mean travel speeds by excluding track regions where prey is not moving (prey displacement is <0.05 mm/sec), and obtained the distribution of mean speeds using kernel density methods (BW = 0.1). This is a skewed distribution, with overall mean prey speed estimated at 0.46 (mm/sec) SE: 0.007, and maximum of 2.5 mm/sec. (**B**) To maintain prey azimuth within the range of observed in successful captures (ie., ±25°), assuming prey is moving at average speed ($\approx$ 0.5 mm/sec), then capture distance needs to be >0.2 mm. This is shown by calculating the changes in θ (dots) between prey moving from initially in front of the larva at a distance $\alpha = r$, with $\beta_0 = 0$ (ie. $\theta = 0$) to different positions $\beta_1$ during a typical interbout interval of 200 ms. The lines show how our linear approximation model, which uses the derivatives evaluated at position $\beta_o$, $r$, capture the growth of $\Delta\theta$ with increasing prey displacement ($\Delta\beta$) and decreasing distance $r$. (**C**) Diagram showing how for prey moving at a fixed speed β the angular velocity of prey azimuth increases with distance

from the larva. Thus, corrections to aiming prior to executing a capture swim become harder when the prey is closer, as they increase with $r^{-2}$ (see text).

We first derive an expression for how sensitive prey azimuth $\theta$ is depending on prey distance $r$, (at some position β), as $\frac{\partial \theta}{\partial r}$. Noting that

$$\alpha = r\cos(\theta) \tag{24}$$
$$\beta = r\sin(\theta) \tag{25}$$
$$\theta = sin^{-1}(\beta/r) \tag{26}$$

we begin can capture the change in angle $\theta$ as the derivative with respect to $r$:

$$\begin{aligned}
\frac{\partial \theta}{\partial r} &= -\frac{\beta}{r^2\sqrt{1-\sin^2(\theta)}} \\
&= -\frac{\beta}{r^2\sqrt{1-\frac{\beta^2}{r^2}}}
\end{aligned} \tag{27}$$

which hints that prey azimuth angle changes inversely proportional with the square distance $r$, and thus for prey located a distance *beta* off the midline axis changes in prey distance, $\Delta \alpha$ can disproportionally effect prey azimuth.

Next, we need to consider changes in azimuth given prey movement , assuming that prey moves a distance $\Delta \beta$ during the inter-bout interval preceding a capture swim. For this we follow a procedure similar to the above to obtain

$$\frac{\partial \theta}{\partial \beta} = \frac{1}{\sqrt{1-\beta^2/r^2}r} \tag{28}$$

shows azimuth is inversely proportional to distance. Using the above derivatives about we can then obtain a linear approximation function for the change in azimuth $\Delta \theta$ against a prey moving a distance β, assuming its initial position is at $(\beta_0, r)$.

$$\Delta\theta(\beta) = \beta\left(\frac{\partial\theta_{(\beta_0,r_0)}}{\partial r}\frac{dr}{db} + \frac{\partial\theta_{(\beta_0,r_0)}}{\partial\beta}\right) \tag{29}$$

where

$$\frac{dr}{d\beta} = \frac{\beta}{\sqrt{\beta^2+r^2}}. \tag{30}$$

*Appendix 5—figure 1B* plots this linear approximation function in comparison to calculated changes using *Equation (26)* within a ranges typical prey travel speeds and capture distances. The linear approximation begins to fail over faster prey speeds at small distances, it nevertheless captures the essential point that over close prey distances $r < 0.2\,\mathrm{mm}$ relatively fast moving prey can cause large azimuth changes of over 40°. Our calculated results are in agreement with prior reports that the angular velocity of prey azimuth monotonically increases from 21° to 67°, between the 1 st and 4th prey-pursuit bout (*Trivedi and Bollmann, 2013*).

Given that in most successful capture events (>75 %) we find prey being between $\theta = \pm25°$ azimuth (see *Figure 6—figure supplement 1*), moving prey at that proximity is most likely difficult to catch and could partly explain the differences in capture success between DF (naive) and LF (experienced) groups. For example, prey moving tangentially, across the horizontal field of view, at a prey distance of 0.2 mm, common to the DF group, with an average prey speed (0.5 mm/sec) (*Figure 1*) will generate a 26° deviation in prey azimuth during a typical inter-bout interval (200 ms); increasing the distance by $\Delta r = 0.25\,\mathrm{mm}$, towards a range typical of LF capture swims, will reduce the deviation in azimuth from 26° down to 12°; If we assume prey is twice as fast, at 1 mm/sec , the above deviations would typically become 45° for DF, and 21° for LF.

Further, the comparison shows that over longer prey distances ($r0.2\,\mathrm{mm}$), the linear function can predict the change in azimuth with approximately less than <5° error, yet this error is magnified to

18°, at $0.1 \nleq r < 0.2$ mm, and to 43° when considering capture distances as close as $r \approx 0.05$ mm. This means that a simple linear control rule that is based on prey-speed could be used to accurately anticipate prey position (see *Bolton et al., 2019*), provided prey is tracked at sufficient distance that exceeds 0.2 mm.

