## [Decision Letter]

Thank you for submitting your article "Learning steers the ontogeny of an efficient hunting sequence in zebrafish larvae" for consideration by *eLife*. Your article has been reviewed by two peer reviewers, and the evaluation has been overseen by a Reviewing Editor and K VijayRaghavan as the Senior Editor. The reviewers have opted to remain anonymous.

The reviewers have discussed the reviews with one another and the Reviewing Editor has drafted this decision to help you prepare a revised submission.

Summary:

The authors analyzed the effects of experience on the kinematics of prey-capture behavior in zebrafish larvae. The researchers used 3 groups: fed with live rotifers, fed with dried food and not fed between 5-7 days after fertilization. Then the prey capture behavior was analyzed in detail. The authors found that LF larva increased the hunt rate compared to the NF and DF larvae, but the hunt duration was reduced. Then the authors found the speed and success rate was increased. Capture-speed and distance to the prey are correlated and the correlation was stronger in LF. Interestingly, the undershooting of initial turns was more frequent in LF. Undershooting and high-speed capture were also correlated. Finally, the authors analyzed the capture efficiency of individual fish, and it was increased in LF. Thus, the authors demonstrated that the initial turn and the hunting sequence were jointly modified by experience and that modification of these components led to capture success.

Essential revisions:

The prey capture behavior of the zebrafish larva has been described in detail including Patterson et al., 2013. In the current manuscript, the authors introduced a new factor in the prey capture behavior, experience, and successfully detected experience-dependent changes in the behavior. From this aspect, the paper describes ethologically interesting and important findings, and the manuscript is written in a very clear manner and easy to follow.

However, the consensus of the reviewers and Reviewing Editor is that the current manuscript needs to move above its current substantially descriptive, albeit valuable, level and some mechanistic insight. These are suggested below, and, as you can see the experiments are feasible. Essentially, the interesting finding, shown in the manuscript – the improved prey-capture skills by feeding on live food – feeds to be further pursued, and the neuronal basis for the improved skills (or at least suggestive results) needs to be presented.

Therefore, the revised version manuscript could include results showing brain functions relevant to the observed behavior. Possibilities include asking which neuronal circuits are involved in the experience-dependent improvement of prey-capture behavior by ablation of specific cell types or sub-circuits of relevant brain areas such as tectum, or detecting changes of neuronal activities caused by learning by calcium imaging, etc. Carefully thought-through experiments, along these lines, will substantially raise the level of the manuscript

1) Some relevant references are missing: Jouary et al., 2016, Johnson et al., 2020 and Mearns et al., 2020 (some of which may be published after submission of the manuscript).

2) Probably an important control will be to expose larvae to live food but both separated by a transparent barrier. This will allow the larvae to interact, recognize but not to be rewarded. If this control will look like exposure to live food, then it will mean that there is no operant learning but rather improvement from repetitive sensory-motor interactions.

3) There are almost no tests for significance for any of the analyses.

4) What's the relative speed of the larvae to the live food? We believe that this is so high that it will almost be the same as the relative speed to the DF. If catching LF and DF requires similar abilities, why then there are such large differences in kinematics between these conditions?

5) Since the analyses were limited for successful hunts, and their rate is different upon the 3 different conditions. Can the different n for all conditions generate any biases?

6) Figure 5B is difficult to read, we would suggest that the dots of the 3 conditions will be on separate panels and a 4th panel with the 3 superimposed curves.

7) Figure 7A, the data points are too big to appreciate their distribution.

8) Do the kinematics of prey-capture behavior differ when the larvae were acclimatized with live prey for shorter (40 min) or longer (2 hrs) periods of time?

9) In the Discussion, the authors suggest that the undershooting in larvae that experience LF may be important for motion paralax. We do not understand how undershooting can contribute to motion paralax as motion paralax measures relative speed and it will not be influenced by the relative final angle with respect to the prey.

10) When humans navigate using a map, they usually use a technique called "offset navigation" For this technique, rather than aiming for the exact position of the target, you would navigate with a slight offset angle (e.g. an offset of 5^o^ to the east), so when you estimate that you are arrived to the expected target and you don't see it, you know that you have to search for it on the west. If you had aimed exactly at the target and you don't see it, you do not know in which direction you have to look for it.

In larvae, undershooting may represent a similar strategy developed upon experience with LF to better estimate the position of the prey after the first J-turn.

11) The finding that prey are mainly detected when they are at a relative angle of 35-50^o^ fits well with the finding in Romano et al., 2015. The latter could be cited to support the results.

12) The second-order conditioning idea developed in the Discussion is a bit complex to understand, needs to be better explained.

---

## [Author Response]

Essential revisions:The prey capture behavior of the zebrafish larva has been described in detail including Patterson et al., 2013. In the current manuscript, the authors introduced a new factor in the prey capture behavior, experience, and successfully detected experience-dependent changes in the behavior. From this aspect, the paper describes ethologically interesting and important findings, and the manuscript is written in a very clear manner and easy to follow.However, the consensus of the reviewers and Reviewing Editor is that the current manuscript needs to move above its current substantially descriptive, albeit valuable, level and some mechanistic insight. These are suggested below, and, as you can see the experiments are feasible. Essentially, the interesting finding, shown in the manuscript – the improved prey-capture skills by feeding on live food – needs to be further pursued, and the neuronal basis for the improved skills (or at least suggestive results) needs to be presented.Therefore, the revised version manuscript could include results showing brain functions relevant to the observed behavior. Possibilities include asking which neuronal circuits are involved in the experience-dependent improvement of prey-capture behavior by ablation of specific cell types or sub-circuits of relevant brain areas such as tectum, or detecting changes of neuronal activities caused by learning by calcium imaging, etc. Carefully thought-through experiments, along these lines, will substantially raise the level of the manuscript

We have now revised the Discussion sections devoted to learning and undershoot behavior in order to improve on mechanistic insights but also their integration with relevant the literature. These revisions then provide the basis for a new section devoted to examining the potential neural circuit mechanisms underlying adaptive hunting behavior. In the new Discussion section, titled "The neural basis of an adaptive hunting routine", we suggest potential circuit mechanism and experimental designs and discuss associated challenges, as requested by the reviewers. To do this we evaluate evidence from published studies and a study co-submitted to *eLife* (Oldfield et al., 2020), which used imaging to pinpoint differences in neural activation patterns between prey experienced and prey naive larvae.

We have also added descriptions of extra revisions to figures and text which we felt could clarify or present further data to strengthen evidence. Some of these revisions were in response to feedback received from other colleagues who read our manuscript.

1) Some relevant references are missing: Jouary et al., 2016, Johnson et al., 2020 and Mearns et al., 2020 (some of which may be published after submission of the manuscript).

Thank you for bringing this to our attention. The submitted version of our manuscript cited the pre-print biorArxiv versions of Mearns et al., 2019, and Johnson et al., 2019, which have now been updated to cite their respective published versions. A citation to Jouary et al., 2016 has now been added to the Discussion section, and particularly in reference to their experimental design and how the undershoot behavior could contribute to distance perception via visual feedback during bout motion.

2) Probably an important control will be to expose larvae to live food but both separated by a transparent barrier. This will allow the larvae to interact, recognize but not to be rewarded. If this control will look like exposure to live food, then it will mean that there is no operant learning but rather improvement from repetitive sensory-motor interactions.

Revealing the learning mechanism and its details would form a very interesting research agenda in moving forward from here, and we have added this point in our revised Discussion. We had made preliminary steps to pursue the barrier idea by constructing arenas with barriers made out of microscopy cover-slips. We first tested whether larvae engage with the prey seen behind the thin glass barrier by comparing hunt initiation in an arena with two glass barriers, one of which was void of prey. We noticed that larvae appear to interact with prey, but have not quantified this, as automatic tracking of eye motion while larvae are touching the barrier was technically challenging. However, we observed that the frequency of such events is low. We believe this was mainly due to the low chances of interacting with prey given that larvae cannot explore the prey occupied space, as in this setup, we are confined to engage only with prey found near the glass when the larva is in proximity. Fixing this by making the prey space depth small, so prey always appears close to the barrier, was technically challenging.

Nevertheless, we did not proceed further with this experimental design because of concerns about the possible effects on learning from the interaction between capture behavior and the barriers. When larvae attempt to capture prey, bumping against the glass-barrier could provide a negative reinforcement, which we cannot control. Given this, we concluded that if this paradigm does not provide evidence that prey exposure is sufficient to improve hunting skills, we would still not be able to exclude it as a hypothesis. This is because we cannot exclude the possibility that a negative reinforcement is delivered when the larva hits against the barrier and this affects the learning process of hunting behavior. In that case, although hunting experience maybe sufficient, and independent of capture outcome, we would not be able to observe such learning, nor to confirm the alternative hypothesis that positive reinforcement is required in order to improve hunting skill.

3) There are almost no tests for significance for any of the analyses.

In analysing major results we have employed a Bayesian inference approach. In the Bayesian paradigm, inferences about the parameters of interest are drawn from the posterior distribution, and testing is optional [1]. This is because instead of testing alternative hypotheses, Bayesian inference allows us to explicitly define statistical models and rigorously evaluate the uncertainty associated in comparing quantities, such as hunt-rates and prey consumption, by inferring full posterior distributions of model parameters from the data. Drawing independent statistical samples of model parameters from these posterior distributions allows us to calculate the probability of a given parameter to be different under two different conditions, thus providing a measure of statistical significance equivalent to the standard p-values obtained when performing statistical tests.

We acknowledge however that in presenting and comparing these parameter distributions in our manuscript, we have not qualified our statements with such probabilities. In this revised manuscript we make better use of the framework and have now thoroughly explicitly added probabilities in figure captions in reference to conclusions (ex. Figure 2 hunt-rates). Also, where we are not doing Bayesian analysis, (ex. see correlations in Figure 4) we have also revised those captions to qualify our statements with significance tests.

Having now introduced probabilities, we are now in position to report that NF larvae are likely to show slightly higher spontaneous eye-vergence rates to DF and LF, something that is in agreement with Filosa et al., 2016. Additionally, we have now extended our results of Figure 2—figure supplement 1, which compared durations of individual hunt episodes. We extend those results by employing Bayesian statistical model inference to estimate the distribution of mean hunt-episode duration. This is accompanied by qualified statements on differences in hunt-episode durations, which here showed that LF larvae are likely to exhibit shorter spontaneous and evoked hunt episodes than controls.

4) What's the relative speed of the larvae to the live food? We believe that this is so high that it will almost be the same as the relative speed to the DF. If catching LF and DF requires similar abilities, why then there are such large differences in kinematics between these conditions?

We agree with the reviewers, that we have not addressed how modifying capture speed and distance could mitigate the challenges faced by larvae in capturing this particular prey type, namely Rotifera Plicatilis. We have now expanded the Discussion on the ontogeny of fast capture swims at increased distance, and also added an analysis in the new appendix 5 to address these points.

Briefly, we argue that here it is not increased capture speed that improves performance per se, but rather that the observed changes in capture distance are the key driver of improved capture success against prey moving at the observed prey speeds. To support this, we present calculations showing that prey-motion during a typical inter-bout interval can induce aiming errors, measured as changes in prey azimuth, that scale with the inverse distance from the prey. We then show that there is +/- 25 degrees optimal prey azimuth that is associated with successful prey capture (new panel in Figure 6—figure supplement 1 ). Given this optimal azimuth range and the observed prey speed range, we find that increasing prey capture distance, within the range found between DF and LF groups, can increase capture success by reducing the chances that faster moving prey move out of the optimal prey azimuth band during the typical inter-bout interval that precedes a capture manoeuvre. Finally, we note that capture speed increases as a consequence of being correlated to distance, most likely so that correct gape-timing is maintained over changing distances. The discussion of these points has now been elaborated and linked with evidence on gape-timing and capture behavior from other studies.

5) Since the analyses were limited for successful hunts, and their rate is different upon the 3 different conditions. Can the different n for all conditions generate any biases?

Such differences in hunt rates could introduce biases if we were to pool data across groups, which we do not do. Nevertheless, it remains something we have been aware during our analysis as differences in hunt-rate are also found within individuals of the same group, as we show there is large variability (see Figure 2 panels A, B, C). For this reason, our conclusions on differences in group behavior are not based on pooled data across individual larvae (see Figure 7), as pooling data to infer mean group behavior can be biased towards very active individuals. Although, intra-group data pooling is used preliminary to reveal differences (ex. undershoot on Figure 5), our outcomes are based on a hierarchical model that estimates group behavior based on the mean behavior of individuals (see Figure 7), and as such it is not influenced by the number of hunt-events generated by each larva. We have now added a note in Figure 5 caption, where we are comparing groups on undershooting using pooled data across individuals, to further stress this point. Additionally, we make also make this point clear in Figure 4, where hunt events are pooled across larva of each group.

Importantly, given our Bayesian approach, differences in n should be reflected in the amount of uncertainty about estimates. However, we have identified that in our 3D model we have failed to reflect the relative uncertainty in estimating group behavior. Differences in uncertainty of estimation between groups is inherently a result of differences in the number of behavior observations obtained from the 60 larvae of each group (some groups generate more hunt events for example). In the previous version of the manuscript we had failed to properly reflect uncertainty, because when estimating group behavior, we excluded larvae on which we had no data on. We have now updated the results of the 3D behavior model to reflect the uncertainty when comparing the behavior between groups of 60 larva for each condition. We have also now qualified observed differences between groups with probabilities.

6) Figure 5B is difficult to read, we would suggest that the dots of the 3 conditions will be on separate panels and a 4th panel with the 3 superimposed curves.

We have now revised this to separate plots from the three groups, as suggested.

7) Figure 7A, the data points are too big to appreciate their distribution.

We have now revised the figure to show higher numbers of smaller sphere as advised. Generally, we found it is difficult to compare distributions in 3D and thus were worried that small spheres would obfuscate the plot. For this reason we had also provided 2D finer grained sections of these distributions in Figure 7—figure supplement 1, and only used the large spheres of the main plot for illustration purposes to show that the groups occupy different positions in parameter space.

8) Do the kinematics of prey-capture behavior differ when the larvae were acclimatized with live prey for shorter (40 min) or longer (2 hrs) periods of time?

Unfortunately this information was not integrated in the dataset, as the acclimatization period did not vary differently between groups and thus it could not add any systematic bias. Nevertheless , identifying the timescales learning to improve hunting skills is indeed a very interesting avenue of further research.

The pace of learning could be associated with the timescales of growth and neural development. We have now included a new figure revealing the correlation of larval size (standard length) against hunting power (HPI) (Figure 3—figure supplement 1). On the one hand, the correlations presented in Figure 3—figure supplement 1 further support that hunting success is not driven by larval growth. This is because although larval growth correlates with HPI in the experienced group (LF), this is not the case for the naive groups (DF,NF). One the other hand, the correlation between larval growth to HPI, unique to the LF group, suggests that hunting skills and growth can be associated. The simplest explanation is off course that efficient hunting skills facilitate growth by increasing food update. However, an interesting possibility is that the timescales of brain development and learning could be related; perhaps because the processes that support physical growth, and depend on food uptake, also support the developmental processes of the brain underlying the learning observed here.

We would also like to add that we do not believe that efficient hunting skills develop rapidly, in the sense that 1 hour extra acclimatisation time, beyond the 40 mins, would not give sufficient exposure to prey such that larvae from the prey-naive groups could match the performance of experienced larvae. This is supported by observing that in Figure 3E none of the naive larvae (DF,NF) managed to reach the level of the most successful hunters (HPI >4) observed in the LF group. This suggests that the kinematics of efficient hunting cannot fully develop in the acclimatization periods provided, something that is also evident in Figure 7B, where there are almost no naive larva that have adopted the kinematics of the most efficient LF larvae (we find 1 DF and 1 NF in the region).

9) In the Discussion, the authors suggest that the undershooting in larvae that experience LF may be important for motion paralax. We do not understand how undershooting can contribute to motion paralax as motion paralax measures relative speed and it will not be influenced by the relative final angle with respect to the prey.

We believe confusion stems from our use of the term motion parallax, which is usually introduced to explain a monocular depth cue that can be used to segment foreground objects from objects further in the background by utilizing the relative velocities of objects moving across the retinae of a moving observer. Although our use here in relation to undershooting is not related to comparing depths between objects in a scene, it nevertheless maintains a reference to utilizing moving retinal projections during a bout of forward motion (visual feedback during motion). We recognized that the idea has not been properly explained in the previous version of manuscript and we have now revised the "Turn undershoot as an active sensing strategy" Discussion section to clarify the mechanism we propose and its predictions. We have also introduced Appendix 5, where we include a schematic to explain the geometry involved and analyse how these could affect the perception of prey motion relative to the larva.

10) When humans navigate using a map, they usually use a technique called "offset navigation" For this technique, rather than aiming for the exact position of the target, you would navigate with a slight offset angle (e.g. an offset of 5^o^ to the east), so when you estimate that you are arrived to the expected target and you don't see it, you know that you have to search for it on the west. If you had aimed exactly at the target and you don't see it, you do not know in which direction you have to look for it.In larvae, undershooting may represent a similar strategy developed upon experience with LF to better estimate the position of the prey after the first J-turn.

We would like to thank the reviewers for pointing us to this interesting alternative explanation. Indeed, we agree such strategy could reduce the uncertainty in directing searches when the target is lost. Although search strategies are known to operate in small brains, as for example is the case for ants when they are not able to locate their nest, it appears zebrafish do not execute searches during hunting, and abandon hunting once a target is out of sight. Thus, we would argue that their hunting behavior does not include heurestics of searching strategies.

11) The finding that prey are mainly detected when they are at a relative angle of 35-50^o^ fits well with the finding in Romano et al., 2015. The latter could be cited to support the results.

Thank you for directing our attention to the relevant data presented in Romano et al., 2015. It is not clear from the data we present whether detection occurs preferentially at those angles, because these relate to detection angles from succesful episodes alone, and not all prey detection events. As such the evidence we present can also be interpreted as prey detected at those angles is more likely to be captured, i.e. prey capture success increases for prey detected at those angles. We discuss, this in reference to data showing that prey is actually detected across a different distribution of angles, which curiously in some studies appears to be unimodal with a peak centered around the frontal visual field as shown in Bolton et al., 2020, Bianco et al., 2011.

Indeed, in Romano et al., 2015 Figure 7C the authors show that prey detection is preferentially triggered over the lateral visual field, with an azimuth distribution that appears very similar to the prey detection azimuth distribution we report in Figure 5—figure supplement 2 over successful episodes alone. This is in agreement with Bianco et al., 2011, for hunting initiation that includes a J-turn but also hunting initiation in partially restrained larvae, but not for freely swimming. Indeed Bianco et al., 2011, notes that prey detection distributions do not match between partially restrained and freely swimming larvae, and thus this points to some limitation the ability of the partially restrained setup to simulate the initial interaction of larvae with prey, and not the ability of larvae to detect prey in their frontal visual field. Yet, in Bianco et al., 2015 Figure 2A a unimodal, frontal visual field, distribution is reported in restrained larva, which now matches the one seen in freely swimming Bolton et al., 2020, Bianco et al., 2011. To verify our distribution of prey detection angle, I contacted Andrew Bolton (Bolton et al., 2020) and he verified that taking the distribution of prey detection angles, conditional on capture success, modifies the unimodal in Bolton et al., 2020 to a bimodal one as in see in our Figure 5—figure supplement 2. Thus, the possibility is raised that although prey can be detected in the frontal visual field robustly, nevertheless prey capture success is associated with prey initially detected at these offset angles. Perhaps part of the differences between prior studies reporting detection angles in unconstrained larvae may relate to whether hunting events are detected using J-turns combined with eye-vergence.

The Discussion section on undershooting has now been revised to include the matters above and now cites data from Romano et al., 2015.

12) The second-order conditioning idea developed in the Discussion is a bit complex to understand, needs to be better explained.

We have now revised that Discussion section to clarify the concepts of 2nd-order conditioning/conditioned reinforcement, as well as added the alternative possibility that undershooting could be reinforced independently of the capture swim. We have also expanding the discussion of this section by adding hypotheses on the behavior adaptation mechanisms. These hypotheses are later used as a basis for our discussion on the potential neural mechanisms of adaptive hunting.

References

[1] C Oldfield, I Grossrubatscher, M Chavez, A Hoagland, A Huth, E Carroll, A Prendergast, T Qu, J Gallant, C Wyart, and E Isacoff. Experience, circuit dynamics and forebrain recruitment in larval zebrafish hunting. eLife, 2020 Submitted.